



# Identification of rock and fracture kinematics in high Alpine rockwalls under the influence of altitude

Daniel Draebing[1,2,3]

[1]Chair of Geomorphology, University of Bayreuth, Bayreuth, 95447, Germany
[2]Derpartment of Physical Geography, Utrecht University, Utrecht, 3584 CB, Netherlands
[3]Chair of Landslide Research, Technical University of Munich, Munich, Germany

*Correspondence to*: Daniel Draebing (d.draebing@uni-bayreuth.de)

**Abstract.** Alpine environments are characterized by fractured rock. Fractures propagate by weathering processes in a
subcritical way, and prepare and trigger rock slope failures. In this study, I investigated (1) the influence of thermal changes
on rock kinematics on intact rock samples from the Hungerli Valley, Swiss Alps. To (2) quantify thermal and ice induced rock
and fracture kinematics and (3) identify differences of their spatial occurrence, I instrumented crackmeters at intact and
fractured rock at four rockwalls reaching from 2585 to 2935 m. My laboratory data shows that thermal expansion follows three
phases of rock kinematics: (1) cooling phase, (2) transition phase and (3) warming phase, which result in a hysteresis effect.
The cooling phase is characterized by rock contraction, while all samples experienced rock expansion in the warming phase.
During the transition phase, rock temperatures differ between rock surface and rock depth, which results in a differentiated
response. The dummy crackmeters in the field reflect temperature phases observed in the laboratory and data suggest a block
size dependency of the transition phase. In fractured rock, fractures open during cooling and reversely close during warming
on daily and annual scale. The dipping of the shear plane controls if fracture aperture decreases with time or increases due to
thermal induced block crawling. On seasonal scale, slow ice segregation induced fracture opening can occur within lithology-
dependent frost cracking windows. Snow cover controls the magnitude and the number of daily temperature changes, reduces
the magnitude of annual cooling but increases the length of the cooling period and, therefore, the potential occurrence of ice
segregation. The effects of snow cover increases with altitude due to longer snow duration. Climate change induced warming
will shift annual thermal stresses at lower altitudes, however, a shortening of the snow period can increase ground cooling and
thermal stress at  higher altitudes but also can reduce the length of the ice segregation period. In conclusion, climate change
will affect and change rock and fracture kinematics and, therefore, rockfall patterns in Alpine environments.

## 1 Introduction

Alpine environments are characterized by high relief due to the interaction of tectonic uplift, climate and erosion (e.g. Schmidt
and Montgomery, 1995; Egholm et al., 2009; Whipple et al., 1999). Tectonics result in the fracturing of rock (Molnar et al.,
2007) which promotes erosion. Glaciers eroded deep Alpine valleys (Harbor et al., 1988; Herman et al., 2015; Prasicek et al.,





2018) and amplified fracturing by thermo-hydro-mechanical rock slope damage during glacial cycles (Grämiger et al., 2017; Grämiger et al., 2018; Grämiger et al., 2020) and internal stress changes following glacier retreat (Leith et al., 2014a, b). Fracturing can be increased by mechanical (e.g. Eppes and Keanini, 2017), chemical (e.g. Dixon and Thorn, 2005) and biological weathering (e.g. Viles, 2012) as well as synergies between weathering processes (Viles, 2013). Therefore, glacial

erosion preconditions and paraglacial processes including weathering prepare and trigger rock slope failures (McColl, 2012; McColl and Draebing, 2019), which are common and hazardous processes (Oppikofer et al., 2008; Krautblatter and Moore, 2014) and key agents of Alpine landscape evolution (Krautblatter et al., 2012; Moore et al., 2009).

Current research highlights the role of mechanical weathering (Eppes and Keanini, 2017). Diurnal and seasonal ambient meteorological changes causing cyclic heating and cooling (Collins and Stock, 2016; Gunzburger and Merrien-Soukatchoff,

2011), wet-dry cycles (Zhang et al., 2015), freeze-thaw cycles (Matsuoka, 2001, 2008) or active-layer thaw (Draebing et al., 2017a; Draebing et al., 2014) produce critical and subcritical stresses that propagate micro-fractures (Draebing and Krautblatter, 2019; Eppes et al., 2018). Several studies investigated the influence of thermal changes on rockwalls and demonstrated the sudden erosion by thermal shock (Collins et al., 2019; Collins et al., 2018) and slow thermal-induced propagation of fractures in Alpine rockwalls (Weber et al., 2017; Hasler et al., 2012; Collins and Stock, 2016) that continuously

weakens rock and can trigger rockfall (Ishikawa et al., 2004; Collins and Stock, 2016). Thermal changes can result in sliding of rock blocks (Gunzburger et al., 2005; do Amaral Vargas et al., 2013). This mechanism can be amplified by rock wedges that fill fractures during cooling phases and thermally expand during warming phases (Bakun-Mazor et al., 2013; Bakun-Mazor et al., 2020).

Freezing of water can cause ice-induced fracture mechanics. Volumetric expansion is a rapid process that occurs when water

freezes to ice and increases in volume by 9 % (e.g. Matsuoka and Murton, 2008). Matsuoka (2001, 2008) observed volumetric expansion induced fracture opening during sudden cooling events in autumn and due to refreezing of meltwater in late spring and early summer. Draebing and Krautblatter (2019) recently simulated this process in the laboratory and showed that volumetric expansion due to refreezing causes subcritical stresses and freezing of saturated fractures are rare but can develop critical stresses. In contrast, ice segregation causes stresses by a thermally induced suction that result in water migration to the

freezing front (Matsuoka and Murton, 2008). This process operates on seasonal scales, can crack intact rocks (Murton et al., 2006) or widen existing fractures (Draebing and Krautblatter, 2019; Draebing et al., 2017b) and cause subcritical stresses (Draebing and Krautblatter, 2019).

While an increased number of studies investigated the kinematics resulting from individual weathering processes, there is still a poor understanding of the spatial variation of these processes. Thermal and ice processes in Alpine environments are

influenced by rock temperature and snow cover that shows high spatial variation in Alpine environments due to altitudinal and topographic effects (e.g. Gruber et al., 2004; Morán-Tejeda et al., 2013; Draebing et al., 2017a). In this study, I collected three rock samples representing different lithologies of Alpine rockwalls in the Hungerli Valley, Swiss Alps, and conducted laboratory tests to investigate (1) the influence of thermal changes on rock kinematics. Furthermore, I instrumented





crackmeters on intact and fractured rock at four rockwalls reaching from 2585 to 2935 m to (2) quantify thermal and ice

induced rock and fracture kinematics and (3) identify spatial differences of kinematics influenced by altitude.

## 2 Research Area

The Hungerli Valley is a hanging valley located in the Turtmann Valley, Valais Alps, Switzerland (Fig. 1a). The geology is
predominantly paragneiss, consisting of schistose quartz slate (Bearth, 1980). At the Rothorn (RH) the schistose quartz slate

is intersected by aplite and at the Hungerlihorli (HH) by amphibolite (Bearth, 1980). The valley was shaped by past glaciations
and regional ice sheet models suggest an ice cover up to 2800 m during Last Glacial Maximum (LGM; Kelly et al., 2004). In
addition, cirque glaciation was abundant during LGM at the Rothorn (RH), Furggwanghorn (FH) and between Hungerlihorli
(HH) and Brändjispitz (BS; Fig. 1c-d). During Little Ice Age, only the Rothorn cirque was ice-covered and is currently
occupied by the remnants of the Rothorn Glacier (RG) that possessed a surface area of 0.053 km² and a length of 405 m in

2011 (Fischer et al., 2014). Rockwalls occur at elevations that range from 2500 up to 3300 m (Fig. 1b). The hanging valley
(Fig. 1d) is occupied by several rock glaciers (Nyenhuis et al., 2005) and talus slopes (Otto et al., 2009). A sediment budget
analysis showed that 1/5 of all stored deposits is derived from rockfall indicating a high rockfall activity (Otto et al., 2009).

## 3 Methods

### 3.1 Laboratory Measurements

To understand controlling factors of rock kinematics, I conducted laboratory measurements on air-dryed rock samples in a
freezing chamber. For this purpose, I collected three approximately 0.4 m long, 0.15 m wide and 0.2 m high rock samples
without visible evidence of weathering from talus slopes below rockwalls with lithologies ranging from aplite (AP, RW-1),
amphibolite (AM, RW-2) to schistose quartz slate (QS, RW-3; in Fig. 1d). I assume that the rock samples are representative

for the rockwall. For a more detailed description of the rock samples and their mechanical properties see Draebing and
Krautblatter (2019). On each rock sample (Fig. 2), two crackmeters were installed at the top (RD1) and at one side of the
sample (RD2) to monitor rock deformation (RD) and rock top temperature (RTT) in 1 min intervals. The two Geokon
crackmeters 4420-3 measure RD and automatically correct thermal expansion of the instrument. After temperature correction,
the resolution of RD is below 0.00075 mm with an accuracy below 0.003 mm, while RTT is measured with an accuracy of

±0.5°C. Two high-precision Greisinger thermistors connected to a Pt 100 temperature sensor (0.03°C accuracy) were used to
monitor rock temperature in the center of the rock sample in 5 cm depth ($RT_{5cm}$) and in 2 cm depth ($RT_{2cm}$). To keep room
temperature constant at low levels, the freezing chamber is located in a fridge that keeps the surrounding temperature at 8 to
10 °C. The freezing chamber itself consists of a custom-made Fryka cooler with a temperature-controlled (0.1°C accuracy)
ventilation system to enable cooling of samples without thermal layering. The rock samples were cooled down from 10-14 °C

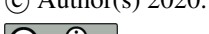



to -8/9 °C $RT_{2cm}$ in 7-9 h. Then, I stopped cooling and enhanced a "natural" warming for 15 to 17 h until 8 °C $RT_{2cm}$ is reached. Based on the linear correlation, I derived the thermal expansion coefficient α:

$$\alpha = \frac{1}{L} * \frac{\Delta RD}{\Delta RT} \tag{1}$$

with L is crackmeter length, ΔRD is rock deformation change and ΔRT is rock temperature change.

## 3.2 Field Measurements

To monitor fracture movement in the field, I installed three 0.4 m long Geokon Vibrating-Wire Crackmeters 4420-1-50 to monitor crack-top temperature (CTT) and crack deformation (CD) with resolution of 0.0125 mm and accuracy of 0.05 mm at RW-1 to RW-3 in 2016 (Fig. 3a-c). Instrumented rockwalls range from 2585 m (RW-3), 2672 m (RW-2) to 2935 m (RW-1) and are exposed in northwest (NW) to northeast (NE) direction (Fig. 1; Table 1). In 2017, three more crackmeters were installed at RW-S at 2723 m at southeast (SE) to northwest (NW) direction (Fig. 3d) to identify effects of different exposition. All crackmeters were fixed by groutable anchors and half tubes protected the devices from snow load (Draebing et al., 2017a; Draebing et al., 2017b). At each rockwall, two crackmeters spanned cracks with apertures that range from 1 to 2 mm at RW-2 to 45 to 135 mm at RW-S (Fig.3; Table 1). In addition, one crackmeter was used as a control crackmeter (Dummy) and was fixed on intact bedrock without any cracks (Fig. 3) to monitor rock deformation (RD) similar to previous studies (e.g. Matsuoka, 2001; Collins and Stock, 2016). Due to snow load damage or technical failures, crackmeters had a different life span and the Dummy at RW-2 failed to record data completely. A 4-Channel Geokon data logger Lc2x4 recorded all data at each rockwall in 3h intervals between 1 September 2016 and mid-August 2017 and in 1h intervals from mid-August 2017 to 31 August 2019. Crackmeter data were temperature-corrected, residual uncertainty quantified and the existence of snow cover was deviated using daily standard deviation of CTT following the technique by Schmid et al. (2012). To analyse the overall fracture movement pattern (Bakun-Mazor et al., 2013), a monthly moving average of CD, RD, CTT and RTT was calculated. The thermal expansion coefficient of the Dummies was calculated based in Eq. 1.

## 3.3 Meteo station data

Air temperature and snow depth data was derived from the MeteoSwiss meteo station Oberer Stelligletscher (2910 m) located 2.5 km SE of the research area in the Matter Valley (MeteoSwiss, 2019a). Probably due to snow cover above 3.5 m, there is a data gap from mid-January to end of February 2018. The temperature data gap was filled using air temperature adapted from near-by meteo station Grächen at 1605 m (MeteoSwiss, 2019b) by applying a linear correlation ($r^2$= 0.85).



## 4 Results

### 4.1 Laboratory measurements

Directly after the start of cooling, crackmeter 1 (C1) at AM and crackmeters at QS revealed a rock expansion, which lasted for a few minutes (initial transition phase; Fig. 4b-c and Fig. 5c, e-f). During cooling, crackmeters at all rock samples showed a rock contraction (cooling phase). The aplite sample revealed a cooling to -16.4 °C RTT at C1 and -18.4 °C RTT at crackmeter 2 (C2) which resulted in a rock deformation of -0.0622 mm at C1 and -0.0572 mm at C2 (Fig. 4a, Fig. 5a-b). Similar to AP, AM experienced rock deformations of -0.0296mm at -16.1 °C RTT at C1 and -0.0360 mm at -16.3 °C RTT at C2 (Fig. 4b,

Fig. 5c-d). QS showed rock deformations of -0.0362 mm at -13.2 °C at C1 and -0.0239 mm at -13.5 °C at C2 (Fig. 4c, Fig. 5e-f). Rock temperature in 2 to 5 cm depth was up to 5 to 7 °C higher than RTT during the cooling phase (Fig. 4 and 5). Based on $RT_{5\,cm}$ measurements using Eq. 1, thermal coefficient ranged from 5.8 ±0.0 $10^{-6}$ °C$^{-1}$ for AM (r²=1) to 7.3 ±0.2 $10^{-6}$ °C$^{-1}$ (r²=0.99) for AP and 7.3 ±0.5 $10^{-6}$ °C$^{-1}$ (r²=1) for QS during cooling.

Several crackmeters (AP C2, AM C2, QS C1 and C2) at all rock samples experienced a sudden rock deformation between +0.004 mm at AP C2 and +0.0135 mm at AM C2 which occurred 9 to 23 min after stopping cooling (transition phase, Fig. 4). At QS, further cooling reversed the rock deformation. The QS sample and AP C1 and AM C1 showed a further rock contraction between -0.0016 mm and -0.0032 mm despite of RTT in the range from -7.3 to -5 °C. In contrast, the closing behaviour corresponded to minimum $RT_{5\,cm}$ that ranged between -9.3 and -8.2 °C (Fig. 5).

Further warming up to 7.7 or 8 °C RTT resulted in rock expansion between 0.0235 mm to 0.0387 mm (warming phase, Fig. 4 and 5). This warming phase induced expansion corresponded to a thermal expansion coefficient of 7.5 ±0.4 $10^{-6}$ °C$^{-1}$ (r²=1) for AP, 7.0 ±0.2 $10^{-6}$ °C$^{-1}$ (r²=1) for AM and 7.1 ±1.7 $10^{-6}$ °C$^{-1}$ (r²=0.99) for QS. All samples showed a hysteresis effect during warming and cooling cycles (Fig. 5), which was amplified using RTT and decreased with further rock temperature depth from $RT_{2\,cm}$ to $RT_{5\,cm}$. All the crackmeters on the top of the sample experienced an anti-clockwise hysteresis effect (Fig. 5a, c, e). A

further closing during the transition phase resulted in different cooling and warming paths. The crackmeter at the side of QS (C2) showed the same hysteresis pattern (Fig. 5f), however, hysteresis pattern of C2 at AP and AM changed from anti-clockwise using RTT to clockwise using $RT_{2\,cm}$ and $RT_{5\,cm}$ (Fig. 5b, d).

### 4.2 Field measurements

#### 4.2.1 Meteorological conditions

At the meteo station at 2910 m, mean annual air temperature (MAAT) ranged from -0.6 °C in 2016/17, to -1.0 °C in 2017/18 and -1.1°C in 2018/19. Daily air temperatures fluctuated between -20 °C and 12 °C (Fig. 6a). After a short period of snow cover with snow depths up to 75 cm between November and December 2016, a second period of snow cover with snow depths up to 150 cm started in mid-January 2017 and lasted until mid-July 2017. Snow onset in the following year was delayed and



started at the end of December 2017. Snow depths reached more than 350 cm and lasted until mid-August 2018. After approximately 2 months without snow, snow cover period started at the end of October 2018 and lasted until the end of July 2019 with snow depths up to 250 cm. Cooling periods lasted from mid-August to mid- January 2017, April 2018 or mid-February 2019 and mean monthly air temperature dropped to -12° C. The warming period lasted 4.5 to 7 months and reached a mean monthly air temperature of 6°C in mid-August.


### 4.2.2 Geologic description of the monitored rock blocks

At RW-1, a 3.5 m long, 1.9 m wide and 2.1 m high aplite block was monitored (Fig. 3a). A slickenslide on top of the block indicates former movement of a previously above-laying block. The monitored block slides on a 20 to 40° inclined shear plane (J3 in Fig. 3a) into the valley with an identical angle than the slickenslide. The block is separated by a 10 to 70 mm wide crack of joint set 1 (J1) from a second 2.5 m long and 2.6 m high block. At RW-2, three blocks were monitored which are incorporated

in a heavily fractured rockwall consisting of amphibolite. All monitored cracks possess an aperture between 1 and 2 mm (Table 1) and are dipping at 50° out of the rockwall (J2), however, the blocks are buttressed by adjacent blocks and the talus slope (Fig. 3b). RW-3 consists of schistose quartz slate and blocks are 0.3 to 1.5 m wide (Fig. 3c). Blocks are separated by 81° inclined and 30 to 50 mm wide cracks (J2), which were monitored. The schist cleavage is dipping at 22° into the rockwall.

RW-S is heavily fractured, which results in the occurrence of a high number of joint sets (Fig. 3d). The monitored blocks dip with an angle of 41° degrees into the rockwall (H2) and crack aperture range from 7.5 to 20 mm at Crack-2 to 45 to 135 mm at Crack-1 (Table1).

### 4.2.3 Field rock deformation measurements

Dummy crackmeters were installed to monitor rock deformation. RTT fluctuated between -15 °C and 10 °C at RW-1 in

2017/18, between -20 °C and 25 °C at RW-3 from 2016 to 2019 and from -15 °C to 25 °C at RW-S from 2017 to 2019 (Fig. 6). The dummy at RW-1 was 204 days covered by snow thereof 21 days lasted the zero-curtain period in 2017/18 (Fig. 6b). In contrast, the snow cover period was reduced to 1 day at the dummy at RW-3 in 2018 (Fig. 6c). The dummy at RW-S experienced 96 days including two days of zero-curtain period and 52 days of snow cover in 2017/18 and 2018/19, respectively (Fig. 6d). All dummies showed small fluctuations of rock deformation that were even reduced when snow cover occurred.

Similar to laboratory experiments, dummies recorded daily and annual cycles of rock expansion during warming and contrary rock contraction during cooling periods (Fig. 6e-g). The thermal expansion coefficients for cooling and warming based on monthly mean RD and RTT was 6.5 $10^{-6}$ °C$^{-1}$ for RW-1 (r² = 0.43), 9.4 $10^{-6}$ °C$^{-1}$ for RW-3 (r² = 0.94) and 9.0 $10^{-6}$ °C$^{-1}$ for RW-S (r² = 0.9). All dummies showed a hysteresis effect, which was amplified at RW-1. The hysteresis effect was characterized by clockwise path at all dummies except RW-S in 2018/19 that experienced an anti-clockwise path (Fig. 6 e-f).






### 4.2.4 Field crack deformation measurements

Crackmeters at RW-1 are located at 2935 m and experienced 192 to 210 days of snow cover at Crack-1 and 215 to 223 days at Crack-2 per year (Fig. 7a). Thereof, annual zero curtain period ranged between 3 and 11 days at Crack-1 and 9 and 39 days at Cack-2. At RW-2 at 2672 m, crackmeters experienced between 69 days (Crack-2) and 73 days (Crack-1) snow cover in

2016/17, thereof between six and eleven days of zero-curtain. In the following years, the snow load damaged the equipment, which resulted in a data gap and incomplete measurement of snow cover duration. RW-3 at 2585 m showed between zero and nine days of snow cover without any zero-curtain period. At the south-facing rockwall RW-S, snow cover was limited to two days at Crack-2 in 2017/18.

On a daily and annual basis, thermal changes results in fracture kinematics. At RW-1, a cooling period lasted from September

2016 to February 2017 and snow onset occurred in mid-November 2016, which decreased daily CTT fluctuations. In the same period, crackmeters experienced an opening up to 0.53 mm at Crack-1 and 0.09 mm at Crack-2 (Fig. 7a). Slow warming started from February to end of May 2017 below snow cover, which resulted in slow crack closing of Crack-1 to 0.48 mm (Fig.7, Fig. 8a-b). In contrast, Crack-2 experienced a slow continuous opening until 0.16 mm. In June, further warming occurred until snow was completely melted and resulted in crack closing to 0.17 mm at Crack-1 and 0.06 mm at Crack-2. Between July and

1 September, CTT showed daily fluctuations and warming, however, crackmeters experienced crack opening to 0.24 mm at Crack-1 and to 0.11 mm at Crack-2, which resulted in an overall irreversible annual opening of the same magnitude (Fig. 8 a-b). The opening pattern during the cooling period repeated itself between September 2017 and December 2017 until snow cover onset. Crackmeters opened up to 0.46 mm at Crack-1 and 0.15 mm at Crack-2. During snow cover until June 2018, crackmeters showed a slow opening to 0.48 mm at Crack-1 and 0.18 mm at Crack-2. In June 2018, the closing pattern repeated

itself during warming below snow cover and cracks closed to 0.30 mm at Crack-1 and 0.11 mm at Crack-2. Crack opening up to 0.36 mm at Crack-1 and 0.23 mm at Crack-2 occurred during the warming period until 1 September. In the year 2017/18 crackmeters experienced an overall irreversible opening of 0.12 mm each. RTT showed a cooling trend until snow onset in December 2018, while Crack-2 experienced a crack opening to 0.31 mm. Crack-1 opened to 0.43 mm in mid-October 2018, the opening pattern amplified until 0.68 mm in December 2018. The closing pattern during warming below snow repeated

itself in June and cracks closed to 0.60 mm at Crack-1 and 0.30 mm at Crack-2. When warming under snow cover amplified, Crack-1 closed to 0.48 mm while Crack-2 opened to 0.33 mm until the snow melted. The following warming period resulted in crack opening to 0.56 mm at Crack-1 and 0.45 mm at Crack-2. The crackmeters experienced an overall annual opening of 0.2 mm at Crack-1 and 0.22 mm at Crack-2 in 2018/19. In total, the block moved between 0.45 to 0.55 mm downslope in three years.

At RW-2, the cooling period from September 2016 to February 2017 decreased CTT to -9.8 °C with crack opening of 0.13 mm at Crack-1 and 0.09 mm at Crack-2 (Fig. 7b). During warming of CTT to -4 °C at Crack-1 and -3.3 °C at Crack-2, Crack-1 and Crack-2 showed further opening to 0.15 mm and 0.20 mm (Fig. 8c-d), respectively. However, further warming of CTT from mid-March 2017 revealed a closing to 0.00 mm at Crack-1 and to -0.09 mm at Crack-2. In 2016/17, Crack-1 showed no





overall crack deformation, while Crack-2 revealed an annual closing of 0.10 mm (Fig. 8c-d). The subsequent cooling period
until November 2017 showed no crack deformation, which was followed by a period of crack opening. In January 2018, snow
cover built up quickly to 350 cm (Fig. 6a) and during warming of CTT, Crack-1 was destroyed at end of April 2019. Crack-2
showed a closing behaviour from mid-May on but was shortly afterwards damaged.

At RW-3, Crack-1 revealed a crack deformation behaviour opposed to CTT with cooling occurred coincidently to crack
opening and warming to crack closing (Fig. 7c).  In contrast, Crack-2 showed no clear crack deformation pattern. During the
cooling period until -9.7 °C CTT in mid-January 2017, Crack-1 revealed a crack opening to 0.17 mm while Crack-2 closed to
-0.28 mm. The following warming period until 9.7 °C CTT in mid-July 2017 showed crack closing to -0.14 mm at Crack-1
and CD fluctuations at Crack-2. In 2016/17, Crack-1 and Crack-2 revealed an overall closing of 0.09 mm and 0.24 mm (Fig.
8e-f), respectively. The following cooling period to -8.8 °C in mid-December 2018 showed an opening up to 0.19 mm at
Crack-1 and nearly no change at Crack-2 (-0.23 mm). During subsequent warming to 10.2 °C in mid-August 2018, cracks
closed to -0.14 mm at Crack-1 and to -0.28 mm at Crack-2. In 2017/18, Crack-1 showed an overall closing of 0.02 mm at
Crack-2 and 0.08 mm at Crack-2. The subsequent cooling period resulted in crack opening to 0.01 mm at Crack-1 and to -0.28
mm at Crack-2 and was reversed to -0.12 mm at Crack-1 and -0.24 mm at Crack-2 during the warming period. In 2018/19,
Crack-1 revealed a CD of -0.13 mm, while Crack-2 showed a CD of 0.07 mm.

At RW-S, cracks showed a closing behaviour during cooling and an opening behaviour during warming (Fig. 7d). During
cooling down to -6.4 °C in December 2017, crackmeters showed a closing to -0.2 mm at Crack-1 and -0.12 mm at Crack-2. In
mid-January 2018, Crack-2 revealed an opening from -0.12 mm to 0.02 mm within 1 month during CTT ranged from -8 °C to
-7 °C (Fig. 8h). This opening continued to 0.04 mm until April when CTT increased to -3.6 °C. Within one month, CTT turned
positive and Crack-2 closed to -0.09 mm. The following warming period showed coincident slight opening of the crackmeters.
In 2017/18, Crack-1 and Crack-2 showed an overall closing of -0.06 mm and -0.09mm, respectively. The following cooling
period with CTT down to -7.2 °C in February 2019 revealed a closing of Crack-1 to -0.24 mm and of Crack-2 to -0.19 mm.
The following warming occurred coincident to crack opening and 2018/19 showed an overall closing of -0.03 mm at Crack-1
and -0.04 mm at Crack-2.

On a daily scale, cooling results in crack opening due to contraction of two rock blocks and warming in crack closing due to
expansion of two rocks (Fig. 9). The peak of opening and closing between individual crackmeters can be different. Crackmeters
at RW-1 experienced daily CTT fluctuations in the range below 10°C and daily CD below 0.1 mm (Fig.10 a-c) during snow-
free periods. The fluctuations were increased up to 16°C and 0.11 mm at RW-2, 23 °C and 0.26 mm at RW-3 as well as 21°C
and 0.32 mm at RW-S.





## 5. Controlling factors of rock and fracture kinematics

### 5.1 Rock kinematics

Laboratory tests showed three phases of rock temperature: (1) cooling phase, (2) transition phase, (3) warming phase with characteristic rock kinematics (Fig. 11a). In addition to these three phases, an initial transition phase occurred when rock samples at room temperature adjust to rapid temperature change at the time the cooling device starts cooling. This phase was also observed in previous laboratory tests by Draebing and Krautblatter (2019), exists due to technical requirements of the cooling process and will not analysed further.

After initial temperature adjustment, the rock samples enter the (1) cooling phase and the data demonstrates that the phase is characterized by rock contraction. Due to the slow conductive heat transport, rock temperature in 2 to 5 cm depth was up to 5 to 7 °C higher than RTT. Thermal coefficients $\alpha$ were calculated based on $RT_{5\,cm}$ measurements using Eq. 1 and $\alpha$ ranged from 5.8 $\pm$0.0 $10^{-6}$ °C$^{-1}$ for AM to 7.3 $\pm$0.2 $10^{-6}$ °C$^{-1}$ for AP and 7.3 $\pm$0.5 $10^{-6}$ °C$^{-1}$ for QS during cooling. These values are in the order of previous thermal expansion coefficients for quartz minerals (Siegesmund et al., 2008) or rock samples (Ruedrich et

al., 2011; Skinner, 1966).

After stopping cooling and enhancing a natural warming of the rock samples, a (2) transition phase starts (Fig. 4). Crackmeters C2 on amphibolite and aplite experienced a sudden rock deformation between +0.004 mm at AP and +0.0135 mm at AM which occurred 9 to 23 min after stopping cooling. The samples rapidly responded to rock warming with thermal expansion, however, thermal expansion especially of AM exceed by far $\alpha$ and was irreversible. Therefore, the data suggest the potential

occurrence of RD induced by thermal shock (Collins et al., 2019; Collins et al., 2018). Despite an increase of RTT from -7.3 to -5 °C, the QS sample and AP C1 and AM C1 showed a further rock contraction between -0.0016 mm and -0.0032 mm. The contraction behaviour corresponds to minimum $RT_{5\,cm}$ that ranged between -9.3 and -8.2 °C (Fig. 5). Therefore, the transition phase is characterized by a temperature difference between rock surface (RTT) and rock depth ($RT_{5cm}$) which is a result of the slow speed of heat conduction. While the rock surface (RTT) is warming and rock is expanding, the overall rock kinematics

is still controlled by cooling of the rock interior and associated contraction.

The following (3) warming phase is characterized by rock expansion (Fig. 4 and 5) which corresponded to a thermal expansion coefficient of 7.0 $\pm$0.2 $10^{-6}$ °C$^{-1}$ for AM, 7.1 $\pm$1.7 $10^{-6}$ °C$^{-1}$ for QS and 7.5 $\pm$0.4 $10^{-6}$ °C$^{-1}$ for AP. Therefore, thermal expansion coefficients are slightly different for cooling and warming phases. In addition to sudden rock deformation during the transition phase, this results in an anti-clockwise hysteresis effect during warming and cooling cycles (Fig. 5). This effect is amplified at

the rock surface (RTT) and decreases with further rock temperature depth from $RT_{2\,cm}$ to $RT_{5\,cm}$ due to slow conductive heat transport. The hysteresis effect was previously observed on limestone, porphyry, tuff and granite (Ruedrich et al., 2011), however, Ruedrich et al. (2011) observed a clockwise hysteresis effect during dry rock conditions, which changed into an anti-clockwise hysteresis in limestone and tuff, when samples were saturated with water.



A subsequent cooling would probably result in a transition phase that would show an opposing behaviour of the observed
transition phase. The phase would be characterized by a delayed response time of rock cooling and ongoing expansion in the
centre of the rock while the rock surface rapidly cools and contracts.

In the field, dummy crackmeters recorded daily and annual cycles of rock expansion during warming and contrary rock
contraction during cooling periods (Fig. 11a). Daily temperature changes affect only the upper 0.5 m of the rock and are
damped with depth (Gunzburger and Merrien-Soukatchoff, 2011), therefore, thermal induced rock kinematics are restricted to
a block size of half a meter. Our data demonstrate that daily thermal cycles are restricted to snow-free periods (Fig. 6b-d) as
previously observed by Draebing et al. (2017b), thus, snow cover reduces the magnitude of thermal changes and thermal
expansion (Fig. 6b-d) due to its insolation properties (Zhang, 2005). On annual scale, mean monthly RD and RTT showed the
same warming and cooling phases as previously observed in the laboratory (Fig. 6e-g). Physical experiments in the laboratory
affect the complete rock sample, therefore, even short-term simulations can be compared to annual temperature changes that
influence complete large rock blocks in the field (Bakun-Mazor et al., 2020). The thermal expansion coefficients for cooling
and warming based on monthly mean RD and RTT range from $6.5 \ 10^{-6} \ °C^{-1}$ for RW-1 (AP, $r^2 = 0.43$) to $9.0 \ 10^{-6} \ °C^{-1}$ for RW-
S (QS, $r^2 = 0.9$) and $9.4 \ 10^{-6} \ °C^{-1}$ for RW-3 (AP, $r^2 = 0.94$). Values of $\alpha$ are in a realistic range (Skinner, 1966), however, the
values are higher than observed $\alpha$ based on calculations with $RT_{5cm}$ in the laboratory. As warming or cooling effects are damped
with rock depth due to heat conduction, lower values of $\alpha$ were expected in the field than in the laboratory. This also results in
low linear correlation between RTT and RD at RW-1, where the effect is increased due to large block size and associated
increased response time to thermal changes. In addition, rockwalls experience frequent moisture changes at the rock surface
during the year (Rode et al., 2016; Sass, 2005), which were not included in the laboratory simulations and could explain higher
observed $\alpha$ in the field. Data from all dummy crackmeters are characterized by a hysteresis effect, which was amplified at RW-
1. Rock blocks at RW-1 are 3.5 m long, 1.9 m wide and 2.1 m high and between 2 and 4 times larger than rock blocks at the
other rockwalls (Fig. 3). A larger rock block size results in a longer heat transfer by conduction and increases the transition
phase, therefore, I interpret the increase of the hysteresis effect as a result from block size and scale effect. The hysteresis
effects at all rockwalls except at RW-S in 2018/19 are characterized by a clockwise path (Fig. 6 e-f), which is in contrast with
my laboratory tests but corresponds with rock behaviour of saturated samples by Ruedrich et al. (2011) and, therefore, could
be result of rock moisture changes.


## 5.2 Fracture kinematics

Rock fracture kinematics are controlled by daily and annual thermal expansion, changes of rock fracture infill and potential
hydrostatic or cryostatic stresses and the dipping direction of the blocks. In fractured rock, cooling results in rock contraction
of intact rocks, however, when neighbouring blocks contract, the in-between fracture opens and contrary closes during
warming induced rock expansion (Fig. 11b; Cooper and Simmons, 1977; Draebing et al., 2017b). This pattern was observed
during daily changes at the rockwalls (Fig. 9) during snow-free periods. My data demonstrate that the number of daily





temperature changes per year is controlled by the duration of snow cover, thus, snow cover isolates the ground (Zhang, 2005) and decreases daily changes (Fig. 10; Luetschg and Haeberli, 2005; Draebing et al., 2017b).

During the absence of snow, thermal cooling of a saturated fracture below the freezing point can result in volumetric expansion

(Matsuoka and Murton, 2008; Walder and Hallet, 1986) which could induce ice stresses in the range of critical cracking (Draebing and Krautblatter, 2019). Laboratory tests by Draebing and Krautblatter (2019) on the identical rock samples showed that water saturation was seldom high enough in artificial 4 mm wide sealed fractures especially in schistose quartz slate, thus, water seepages via micro-fractures. Therefore, a saturation of larger fractures as observed in the field (Fig. 3) is unrealistic. Matsuoka (2001, 2008) observed sudden short-term opening of rock wedges by volumetric expansion, however, monitored

blocks and fractures were smaller and therefore a saturation above the threshold of 91 % (Walder and Hallet, 1986) is more likely. The data of Crack-1 at RW-1 experienced a sudden opening in October 2016 and 2017, when CTT turned negative, however, the fracture is 2.1 m high, 1.9 m wide and between 10 and 70 mm wide (Fig. 3 a), therefore a preconditioning saturation is impossible and a volumetric induced fracture opening highly unlikely.

On annual scale, rock temperature fluctuates sinusoidal and can be differentiated into a cooling and warming phase. The

cooling period at RW-1 lasted usually from September until the onset of snowfall in November to February, when snow depth was thick enough to isolate the ground (Draebing et al., 2017a; Haberkorn et al., 2015a; 2015b). However, crackmeters at RW-1 experienced a cooling below snow cover from September 2016 to February 2017. Meteo station data revealed a snow depth up to 75 cm thick from November 2016 on, which increased up to 150 cm from mid-January 2017 on. I interpret the observed cooling as the thin snow cover effect that increases thermal gradients and therefore amplifies cooling by conductive heat

transport as previously reported by Keller (1993) and Phillips (2000). In general, the cooling phase resulted in fracture opening up to 0.53 mm at Crack-1 and 0.09 mm at Crack-2 in 2016/17, 0.22 mm at Crack-1 and 0.04 mm at Crack-2 in 2017/18 and 0.32 mm at Crack-1 and 0.08 mm at Crack-2 in 2018/19 (Fig. 7a). I interpret the opening behaviour as a result of thermal contraction. The observed pattern is consistent to previous studies of fracture movement at the Matterhorn (Hasler et al., 2012; Weber et al., 2017).

At lower altitudes, snow onset occurred already during the warming phase. Rock cooling resulted in crack opening of 0.13 mm at Crack-1 and 0.09 mm at Crack-2 in 2016/17 at RW-2 (Fig. 7b). At RW-3, Crack-1 experienced an opening of 0.17 mm in 2016/17, 0.33 mm in 2017/18 and 0.15 mm in 2019/19 (Fig. 7c). In contrast, neighbouring Crack-2 showed a closing of 0.29 mm in 2016/2017 and no closing or opening pattern in subsequent years. At RW-S, all crackmeters show a closing trend during the cooling period that ranged between 0.2 mm in 2016/17 and 0.18 mm in 2018/19 at Crack-1 and 0.12 mm in 2016/17

and 0.1 mm in 2018/19 at Crack-2 (Fig. 7d). The data suggest that the contrary behaviour is associated to dipping direction of blocks and will be discussed at a later stage.

Warming periods usually started between December and February and revealed thermal- and ice-induced kinematics. At RW-1, the warming period was characterized by a slow warming below snow cover that was amplified in June. The slow warming resulted in slight crack closing of 0.05 mm in 2017, 0.02 mm in 2018 and 0.08 mm in 2019, which corresponded to thermal

expansion of rock (Fig. 7a). Meteo station data at 2910 m in flat terrain showed that snow depth decreased significantly from





June on (Fig. 6a). Therefore, I interpret the amplified warming at RW-1 as a result of increased heat transport due to a settling and melting snow cover. The warming induced fracture closing ranged from 0.17 mm in 2017, 0.18 mm in 2018 to 0.15 mm in 2019 at Crack-1. At RW-3, warming results in closing of Crack-1 of 0.31 mm in 2017, 0.33 mm in 2018 and 0.13 mm in 2019. In contrast, Crack-2 at RW-3 shows no clear pattern. Similarly, Crack-1 at RW-S experienced an opening of 0.11 mm
in 2018 and 0.1 mm in 2019 and Crack-2 showed a slight opening in 2019. All blocks neighbouring the cracks are dipping into the rockwall, which suggest that crack kinematics are primarily controlled by the dipping direction.

Several crackmeters experienced a slow ice-induced crack opening (Fig. 11c). At RW-1, Crack-2 showed a slow continuous opening of 0.07 mm in 2017 and 0.03 mm in 2018 (Fig.7a). The opening occurred over 5 to 5.5 months in a mean monthly CTT range between -9 to -2 °C in 2017 and -8 to -1 °C in 2018 (Fig. 8b). At RW-2 a similar slow 1.5 to 2 month lasting crack
opening in the range of 0.02 mm at Crack-1 and 0.11 mm at Crack-2 was observed in 2017 (Fig. 7b). The opening occurred in a mean monthly CTT range between -5 to -4 °C at Crack-1 and -5 to -3.3 °C at Crack-2 (Fig. 8c-d). The opening pattern seems to repeat itself in the following year but crackmeter failure due to snow load prevents an identification. At RW-S, Crack-2 experienced a 0.14 mm opening in mid-January 2018 that lasted 1 month at a mean monthly CTT between -8 and -7 °C (Fig. 7d, Fig. 8 h). The fracture continued to open by 0.02 mm in 2.5 month until a mean CTT of -3.6 °C was reached (Fig. 8h).
After exceeding a CTT between -4 and -1 °C, all cracks experienced a closing behaviour. The slow speed and long duration of the fracture opening process in combination with the CTT range suggest the occurrence of ice segregation that induced the observed fracture opening. Ice segregation is a slow process driven by temperature-gradient induced suction (Matsuoka and Murton, 2008; Williams and Smith, 1989). Cracks can infill with ice by refreezing of meltwater, ice infill growth slowly with time by cryosuction (Weber et al., 2018; Draebing et al., 2017b) and build up ice pressure and stresses in a subcritical range
(Draebing and Krautblatter, 2019) which induce slow fracture opening (Draebing and Krautblatter, 2019; Draebing et al., 2017b). The observed crack opening occurred in a temperature range similar to the frost cracking window between -8 to -3 °C suggested by Anderson (1998) and in the temperature range of enhanced frost cracking between -6 to -3 °C observed at laboratory tests on Berea sandstone by Hallet et al. (1991). Numerical models suggest that the lower limit of frost cracking by ice segregation depends on water permeability and the upper boundary depends on rock strength (Walder and Hallet, 1985;
Rempel et al., 2016). Therefore, ice segregation and induced fracture opening is rock type dependent, which is supported by laboratory tests by Murton et al. (2006) and Draebing and Krautblatter (2019). Draebing and Krautblatter (2019) tested ice segregation induced fracture opening on rock samples from the rockwalls investigated in this study and observed an upper boundary of frost cracking of -2.35 °C for AP (RW-1), -0.64°C for QS (RW-3 and RW-S) and -0.04°C for AM (RW-2). These values slightly differ from temperature ranges observed at RW-1, RW-2 and RW-S, which can result from limited water
availability in the field, while water availability was not limited in the laboratory study, and lower rock strength of the samples compared to rockwall strength. Exceeding the temperature threshold results in slow fracture closing due to ice relaxation and an ice pressure reduction (Draebing and Krautblatter, 2019; Draebing et al., 2017b).

Following snowmelt, rock temperature at RW-1 adjusts to atmospheric conditions. Crackmeters experienced a crack opening after complete snowmelt between July and 1 September despite a temperature warming. The opening ranged between 0.06 mm



in 2019 and 0.24 mm in 2017 at Crack-1 and 0.11 mm in 2017 and 0.12 mm in 2018 at Crack-2. This behaviour is similar to the transition phase observed in the laboratory. The rock surface is warmed, however, the temperatures at depth of the large blocks are not affected by warming yet due to slow conductive heat transport and the block size results in an increased response time of kinematics to warming.

The overall and annual movement seems to be controlled by dipping of the blocks and seasonal thermal- and ice-induced

kinematics. Rock blocks at RW-1 are large and dip out of the slope (Fig. 3a). The observed block experienced rock creeping that results in a crack opening of 0.46 mm at Crack-1 and 0.15 mm at Crack-2 in 2016/17, 0.12 mm at each crackmeter in 2017/18 and 0.2 mm at Crack-1 and 0.22 mm at Crack-2 in 2018/19. In total, the block moved between 0.45 to 0.55 mm downslope in three years. The opening pattern is not restricted to summer periods as observed at Matterhorn (Hasler et al., 2012; Weber et al., 2017) where the authors explained this opening by hydro-thermal induced strength reduction. I interpret

the observed movement as a combination of thermal induced rock "crawling" previously observed by Gunzburger et al. (2005) amplified by ice segregation below the block recorded at Crack-2 (Fig. 11d). The occurrence of a slickenslide on top of the block indicates former movement of a previously above-laying rock, therefore, this rock movement is not a singularity in the Rothorn cirque.

The reduced time series at RW-2 enables only the analysis of rock movement in 2016/17. Despite all blocks are dipping at 50°

out of the rockwall (J2), Crack-1 showed no overall crack deformation and Crack-2 revealed an annual closing of 0.10 mm (Fig. 8c-d). This is due to buttressing by adjacent blocks and the talus slope (Fig. 3b), which control and reduce the movement. In contrast, all cracks probably experienced ice-induced opening on seasonal scale, which was reversed by subsequent ice relaxation induced crack closing.

At RW-3, crackmeters reveal overall crack closing of 0.09 mm at Crack-1 and 0.24 mm at Crack-2 in 2016/17, 0.02 mm at

Crack-1 and 0.08 mm at Crack-2 in 2017/18 and 0.13 mm at Crack-1 in 2018/19. This can be explained by the schist cleavage, which acts as a shear plane and is dipping at 22° into the rockwall. At RW-S, the monitored blocks dip with an angle of 41° degrees into the rockwall (H2) and cracks showed an overall closing of 0.06 mm at Crack-1 and 0.09 mm of Crack-2 in 2017/18 and 0.03 mm at Crack-1 and 0.04 mm at Crack-2 in 2018/19. The closing is controlled by the dipping of the shear plane which increases stability (Fig. 11e; Selby, 1980). The opposing behaviour of Crack-2 at RW-3 with crack opening of 0.07 mm in

2018/19 can be a result from kinematics of larger neighbouring block that prevented fracture closing similar to wedge effects observed by Bakun-Mazor et al. (2020).

## 5.3 Altitudinal effects on rock and fracture kinematics and influence on rockfall

High mountain topography can result in strong spatial variability of snow cover and rock surface temperature due to elevation

differences, geometry and aspect, which influences thermal and ice induced rock and fracture kinematics. The snow cover period declined with decreasing altitude (Fig. 6 and Fig. 7) from RW-1 to RW-3 from 192 to 223 days (RW-1) to 0 to 9 days (RW-3). Due to snow load damage, the time series of RW-2 is too small to compare it with RW-S to identify the influence of aspect on snow duration (Fig. 7). During the snow-free period, the number of daily temperature changes and daily crack





deformation decreased with longer snow cover on daily scale. The number of daily temperature changes decreased up to 55
to 60% with altitude in the Hungerli (Fig. 10), while the magnitude of daily CTT changes was reduced by up to 10 °C and,
therefore, daily CD decreased with elevation. An influence of aspect on daily CD was previously observed by Draebing et al.
(2017b), however, a clear pattern cannot be distinguished in the Hungerli data. Data from RW-S (Fig. 10j-l) showed lower
daily CTT change than 150 m lower located RW-3 (Fig. 10g-i), however, daily CD was increased at RW-S despite identical
lithology. This effect was even amplified compared to RW-2 (Fig. 10d-f), however, the overlapping data period is too short to
draw conclusions. Daily CTT changes only influence near surface areas lower than 0.5 m depth (Gunzburger and Merrien-
Soukatchoff, 2011) and, therefore, the increase of number and magnitude of daily temperature changes and associated crack
deformation with altitude can result in higher frequency of small-scale rockfall at lower elevations.

The timing of snow cover onset controls ground cooling, which is decreased at higher altitudes due to earlier onset and probably
thicker snow depth. Snow cover onset between November and December at RW-1, while snow onset was delayed between 2
to 3 weeks in 2018/19 to 1 month in 2017/2018 and 3 month in 2016/17 at RW-2. This delay was even more amplified at
rockwalls at lower elevations (e.g. RW-3). The early onset of thick snow cover at RW-1 prevents or damps further ground
cooling (Luetschg and Haeberli, 2005; Draebing et al., 2017a), which can occur at lower elevation rockwalls. If snow depth is
below a certain threshold (Keller, 1993; Phillips, 2000), thin-snow cover effect can enhance cooling as observed at RW-1 in
2016/17, when ground continued to cool despite existent snow cover. The meteo station recorded up to 75 cm snow depth,
however, the meteo station is located on flat terrain and snow depth is not transferable to rockwalls due to different wind
patterns, slope morphometry and roughness (Haberkorn et al., 2015a; Phillips, 2000; Wirz et al., 2011). The damping of ground
cooling resulted in a small difference of CTT at all rockwalls. The data demonstrates differences of minimum mean monthly
CTT between each years but the cooling pattern is not differing between rockwalls and reached monthly CTT between -10 and
-7° C (Fig. 8). Potential ice segregation started during warming between -9 and -4°C, therefore the onset of ice segregation
induced fracture infill seems not to be affected by the decrease of ground cooling. The delayed response is probably a result
of ice infill that needs to establish by refreezing (Draebing et al., 2017b; Weber et al., 2018) before sufficient cryosuction can
build-up ice pressure induced fracture opening. However, a longer snow period at higher location results in a longer time
period of negative CTT which can prolong to duration of ice segregation as previously suggested by Draebing et al. (2017b).
Therefore, ice induced kinematics are increased at higher elevations and can more likely prepare and trigger rockfall. During
warming, mean monthly CTT was increased with decreasing altitude (Fig. 8). In combination with damped ground cooling at
higher elevations, this resulted in an increased annual warming and cooling cycle at lower elevation, which can result in higher
thermal stresses at lower elevations and more thermal-induced rockfall.

Climate change will affect rock temperature and snow cover and, therefore, will change rock and fracture kinematics in high
Alpine rockwalls. Snow cover and snow duration is expected to decrease, while ground temperature will increase due to climate
change (Gobiet et al., 2014; Bender et al., 2020). A decrease in snow cover and duration can enhance ground cooling at higher
locations, however, the period of potential ice segregation will be reduced. Due to longer snow-free periods, the number and
potentially the magnitude of daily thermal changes and daily CD can increase. An increase of rock temperature will only shift

the annual warming and cooling cycles but probably not increase the temperature range of annual changes and thermal stresses. Therefore, climate change will especially influence high altitude rockwalls, where the occurrence of ice-induced stresses will
probably be reduced, while the occurrence of daily thermal stresses will be increased.

## 6. Conclusion

Laboratory tests on intact rock samples show three phases of temperature development: (1) cooling phase, (2) transition phase and (3) warming phase. The cooling phase is characterized by rock contraction. During the transition phase, rock temperatures
increase at the rock surface but the rock block is still cooling in the centre, which results in overall rock contraction. In the warming phase, all samples experienced rock expansion. A further transition phase can be expected, when temperatures switch from warming to cooling. The transition phases and potential irreversible rock deformation by thermal shock result in an anticlockwise hysteresis effect. The dummy crackmeters in the field showed identical temperature phases with characteristic rock deformation but hysteresis effect followed more a clockwise path. The transition phase was increased at larger blocks due
to longer response time of conductive heat transport. Thermal induced rock fracture kinematics showed a fracture opening during cooling and reversed closing during warming on daily and annual scale. If the shear plane dips into the rockwall, fracture aperture decreases with time, while dipping out of the rockwall can result in thermal induced block crawling. On seasonal scale, slow ice segregation induced fracture opening can occur below and without snow cover within lithology-dependent frost cracking windows. The snow cover period decreases with decreasing altitude, snow cover reduces significantly the magnitude
and the number of daily temperature changes and, therefore, daily induced thermal stresses. On annual scale, a thick snow cover reduces the cooling of rock at higher altitudes, which decreases annual thermal stresses, however, snow cover prolongs the period of negative rock temperatures where ice segregation can occur. Due to climate change, warming will shift annual thermal stresses at lower altitudes, however, a shortening of the snow period can increase ground cooling and thermal stress but also reduces the ice segregation period at higher altitudes. In summary, climate change will affect and change preparatory
and triggering factors of rockfall.

**Data Availability.** All data is available in the Supplementary Information.

**Author Contributions.** DD designed the research, installed the crackmeters, processed the data and wrote the manuscript.


**Competing Interests.** The author declares that he has no conflict of interest.

**Acknowledgements.** This study was funded by German Research Foundation (DR1070/1-1). Laboratory and fieldwork support by Till Mayer was highly acknowledged. Furthermore, the author thanks Samuel McColl, Arne Brandschwede, Florian



Strohmaier, Simon Schäffler, Arne Thiemann, Ann-Kathrin Baßler, Alexandra Weber, Veronika Stiller and Helle Nannestad
for fieldwork support and Michael Krautblatter and Maarten Kleinhans for hospitality.

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



**Figures:**

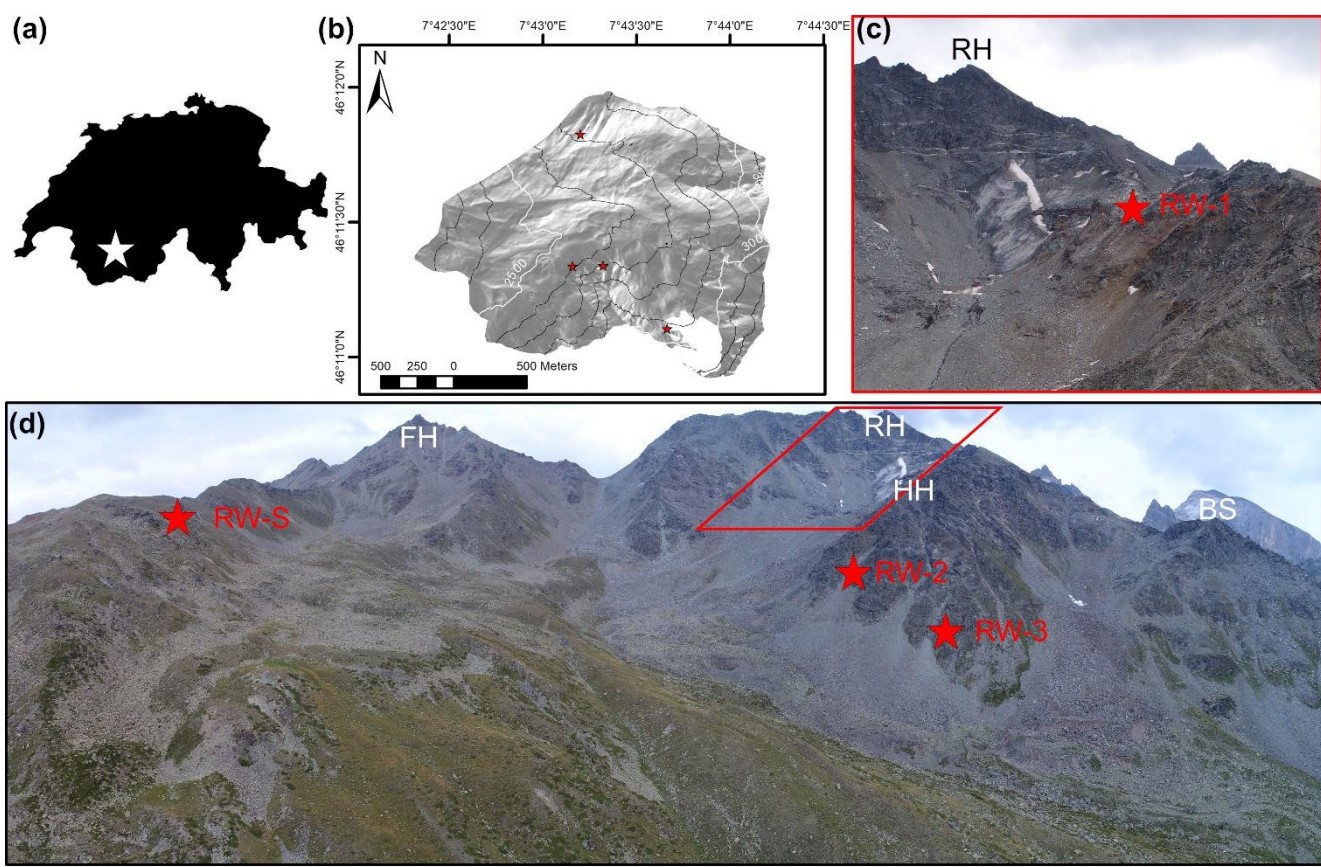

**Figure 1: (a) Location of the Hungerli Valley in Switzerland. (b) Hillshade model with highlighted crackmeter locations (red stars; Swiss Alti3D 2 m provided by the Federal Office of Topography, swisstopo). (c) Drone photo of the Rothorn cirque with location of RW-1. (d) Drone photo of the Hungerli Valley with locations of RW-2, RW-3 and RW-S. The peaks Furggwanghorn (FH, 3161 m), Rothorn (RH, 3277 m), Hungerlihorli (HH, 3007 m) and Brändjispitz (BS, 2852 m) are highlighted.**

Earth **Surface**
Dynamics
Discussions


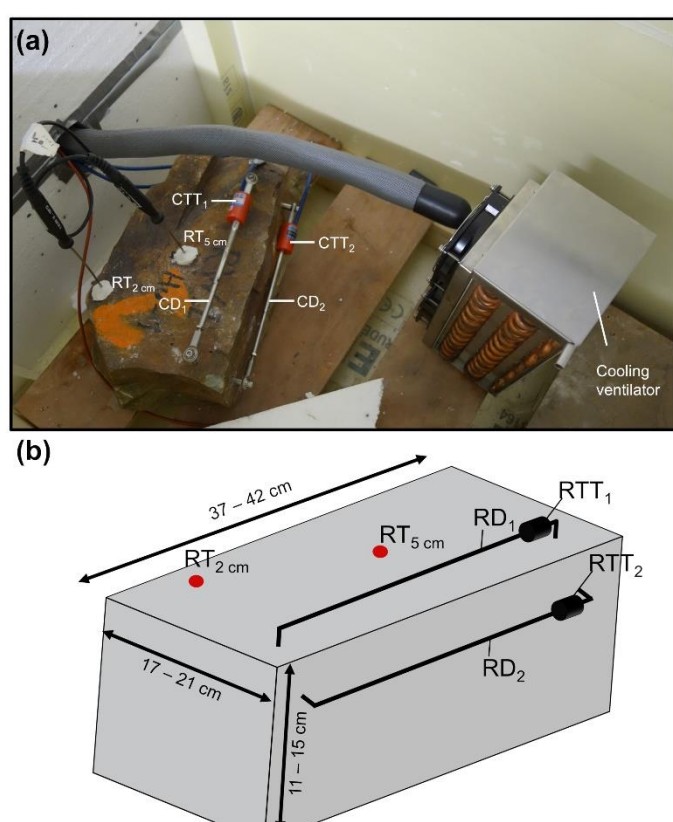

Figure 2: (a) Cooling chamber with aplite sample from RW-1. (b) Schematic illustration of the sensors at the rock samples.





**Figure 3: (a) Three crackmeters were attached on two large (>2 m) aplite blocks at RW-1 at 2935 m. (b) At RW-2 at 2672 m, two crackmeters were installed on densely fractures amphibolite blocks. Crackmeters monitor CD and RD of schistose quartz slate (c) at RW-3 at 2585 and (d) at RW-S at 2723 m.**



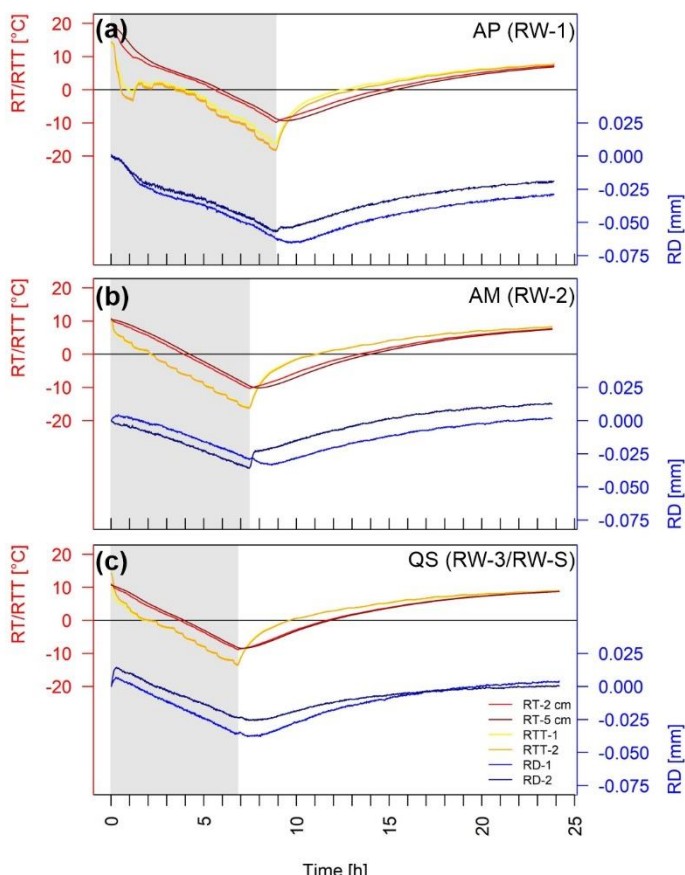

**Figure 4: Laboratory crackmeter measurements of rock samples. Rock-top temperature (RTT), rock temperature (RT) and rock**
**675  deformation (RD) of crackmeter 1 and 2 plotted versus time for (A) the aplite (AP) sample from RW-1, (b) the amphibolite (AM)**
**sample from RW-2 and (c) the schistose quartz slate (QS) sample from RW-3 which is also representative for RW-S.**



**Figure 5: Rock deformation plotted versus rock-top temperature or rock temperature for (a) crackmeter 1 and (b) crackmeter 2 of the aplite sample (RW-1), for (c) crackmeter 1 and (d) crackmeter 2 of the amphibolite sample (RW-2) and for (e) crackmeter 1 and (f) crackmeter 2 of the schistose quartz slate sample (RW-3 and RW-S).**



**Figure 6: (a) Air temperature and snow depth from meteo station Oberer Stelligletscher plotted for the period from September 2016 to August 2019. (b) Rock-top temperature, monthly mean rock-top temperature, rock deformation and monthly mean rock deformation for the dummy crackmeters at (b) RW-1, (c) RW-3 and (d) RW-S. Light-grey rectangles indicate the snow cover period and dark grey rectangles the zero curtain period. Monthly mean rock deformation plotted versus monthly mean rock-top temperature for (e) RW-1, (f) RW-3 and (g) RW-S. Red rectangles indicate start, red dots the beginning of new measurement period and red triangles the end of measurements. Colour of graphs indicate data from 2016/17 (lightblue), from 2017/18 (blue) and 2018/19 (green).**





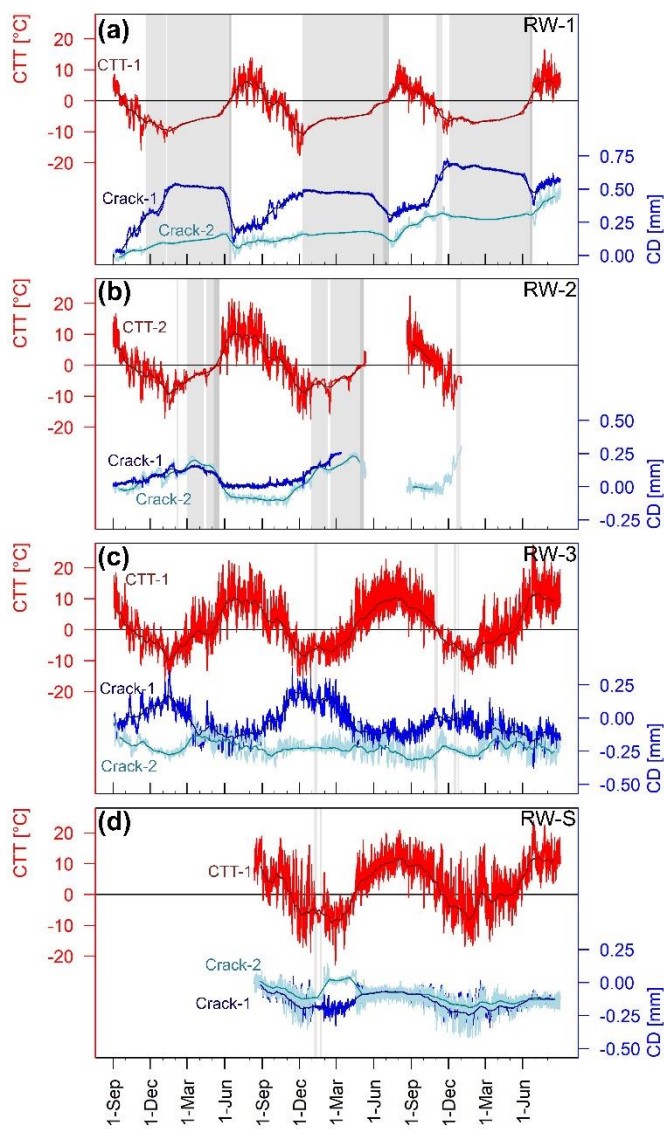

Figure 7: Crack-top temperature, monthly mean crack-top temperature, crack deformation and monthly mean crack deformation for the crackmeters at (a) RW-1, (b) RW-2, (c) RW-3 and (d) RW-S. Light-grey rectangles indicate the snow cover period and dark grey rectangles the zero curtain period.



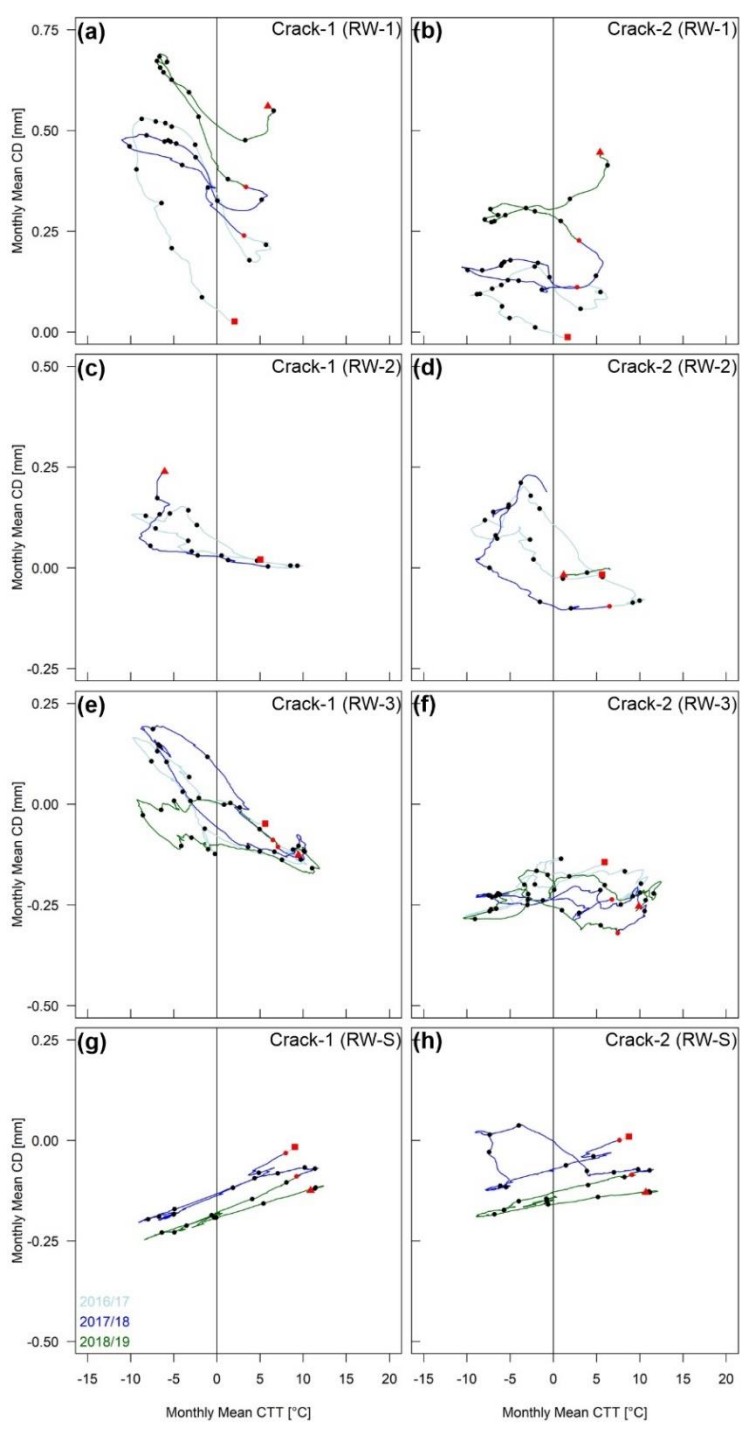

**Figure 8: Monthly mean crack deformation plotted versus monthly mean crack-top temperature for (a) crackmeter 1 and (b) crackmeter 2 at RW-1, for (c) crackmeter 1 and (d) crackmeter 2 at RW-2, for (e) crackmeter 1 and (f) crackmeter 2 at RW-3, and for (g) crackmeter 1 and (h) crackmeter 2 at RW-S. Red rectangles indicate start, red dots the beginning of new measurement period and red triangles the end of measurements. Colour of graphs indicate data from 2016/17 (lightblue), from 2017/18 (blue) and 2018/19 (green).**





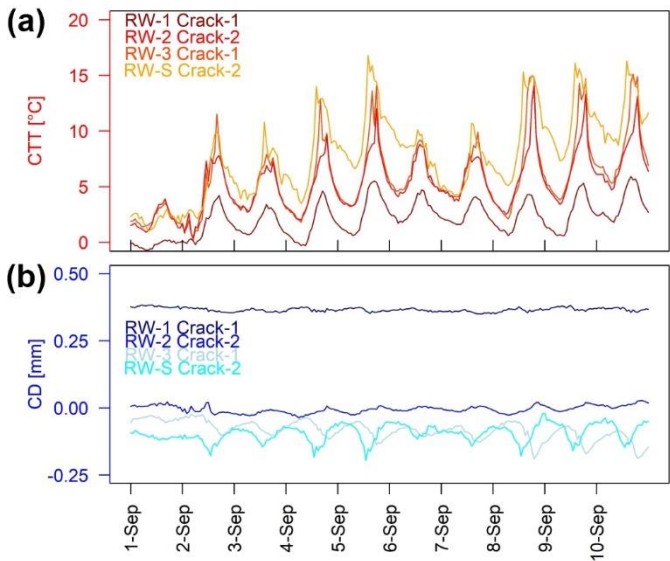

705

**Figure 9: (a) Crack-top temperature and (b) crack deformation for selected crackmeters of RW-1, RW-2, RW-3 and RW-S between 1 September 2018 and 10 September 2018.**





**Figure 10: Daily crack deformation plotted versus daily crack-top temperature for RW-1 in (a) 2016/17, (b) 2017/18 and (c) 2018/19, RW-2 in (d) 2016/17, (e) 2017/18 and (f) 2018/19, RW-3 in (g) 2016/17, (h) 2017/18 and (i) 2018/19, and RW-S in (j) 2016/17, (k) 2017/18 and (l) 2018/19.**





**Figure 11: Conceptual illustration of (a) thermal-induced rock kinematics, (b) thermal-induced fracture kinematics, (c) ice-induced fracture kinematics, (d) thermal-induced rock crawling on a shear plane dipping out of the rockwall and (e) thermal-induced rock kinematics on a shear plane dipping into the slope. Insets in (a) and (b) show daily temperature oscillation below snow cover and without snow cover.**





**Tables**

**Table 1: Field crackmeters.**

| Rockwall/ Crackmeter | Elevation [m] | Aspect [°] | aperture [mm] | snow cover duration [d] | | | zero curtain duration [d] | | |
|---|---|---|---|---|---|---|---|---|---|
| | | | | 2016/17 | 2017/18 | 2018/19 | 2016/17 | 2017/18 | 2018/19 |
| *RW-1* | *2935* | | | | | | | | |
| Crack-1 | | 302 | 10-70 | 192 | 208 | 210 | 3 | 11 | 3 |
| Crack-2 | | 20 | 3-35 | 221 | 223 | 215 | 9 | 39 | 10 |
| Dummy | | 312 | | | 204 | | | 21 | |
| *RW-2* | *2672* | | | | | | | | |
| Crack-1 | | 300 | 2 | 73 | 67* | 8* | 6 | 7* | |
| Crack-2 | | 298 | 1-2 | 69 | 109* | | 11 | | |
| *RW-3* | *2585* | | | | | | | | |
| Crack-1 | | 288 | 45-50 | 0 | 5 | 9 | 0 | 0 | 0 |
| Crack-2 | | 282 | 30-40 | 0 | 0 | 1 | 0 | 0 | 0 |
| Dummy | | 307 | | 0 | 1 | 1 | 0 | 0 | 0 |
| *RW-S* | *2723* | | | | | | | | |
| Crack-1 | | 169 | 45-135 | | 5 | 0 | | | 0 |
| Crack-2 | | 283 | 7.5-20 | | 21 | 0 | | 2 | 0 |
| Dummy | | 202 | | | 96 | 52 | | 2 | 0 |

* incomplete data set due to snow load damage

720