# Peer review of "Identification of rock and fracture kinematics in high alpine rockwalls under the influence of elevation"

_Earth Surface Dynamics, 2020_

## Referee Comment (RC1) · Anonymous Referee #1 · 19 Oct 2020

Review of Draebing, "Identification of rock and fracture kinematics in high Alpine rockwalls under the influence of altitude" submitted to Earth Surface Dynamics.

General Comments

This manuscript presents a highly detailed study of the cooling and heating effects of alpine rocks in a laboratory and field setting. The author should be commended for capturing this type of data set as it is clear that it has the ability to inform on several key attributes of the geomorphological behavior of high altitude rock walls. The author makes several interesting points and clearly has collected a significant data set to stand behind some important interpretations. The confirmation of hysteretic thermally-generated fracture deformation is a very good contribution, as is the observation that the thermal expansion coefficients are different between cooling and warming. In addition, the observations that snow cover (and lack thereof from future climate change) will affect the cyclic thermal effect at different altitudes is an important result in itself, especially since it is clearly shown by the field data. However, overall, I found the structure and presentation of the manuscript requiring significant revision. At the outset, the Abstract should summarize the main findings of the research. I found these difficult to identify. The Abstract discusses rock fracture, altitudinal dependence, and climate change, but also presents results on shear plane controls, ice segregation effects, and snow cover. Whereas these latter three items are all part of the overall arching theme of rock fracture and altitudinal dependence, I believe that setting up the subject more clearly would assist with the delivery of the major findings. That is, I would suggest text along the lines of: "In this study, I investigate the various altitudinal effects (i.e., thermal cycling, snow cover, ice segregation) on rock fracture and place these in the context of climate change." On this last notion, climate change is discussed in the manuscript, but really only casually at the end and I feel that to tackle this subject, the Discussion should include a more in depth overview of what others have found on this subject. Referring back to the Abstract, I noted that the Conclusions section is almost exactly the same as the Abstract. Please note that the Abstract should provide an overview for readers who have not yet read the paper. The Conclusion summarizes the work for those readers who have finished reading the paper. They should therefore not contain the same identical content. Finally, regarding the overall structure of the manuscript, I found the Results and parts of the Discussion section quite tedious within the overall presentation. Whole sections of the text could be summarized in a table or chart, and it is not necessary to highlight the results of every fracture measurement or temperature reading. I therefore recommend a rewrite of large sections of the text (including the Abstract, Results, Discussion, and Conclusion) so that the results and main findings of the research are more clearly represented. Overall, I found the manuscript did not clearly deliver the apparent intended results of the study.

Specific Comments

Dummy Crackmeters. The use of the word "Dummy" should probably be changed to "Control". Dummy implies that the device does not provide any useful information. Regarding the devices themselves, they are not quite control crackmeters either, however, since they are still measuring rock deformation, just not across a fracture. Thus, whereas they provide some guidance for understanding the fracture measurements, there use should be put into context that they are better for understanding the non-fractured behaviour of the rock mass. Also, since the crackmeters were temperature corrected (L113), it was unclear how the controls were used for verification of the measured signals.

Thermal Shock. The term thermal shock is used to describe rapid deformation and fracture observed in several previous studies (L265), however my understanding is that those studies did not depend on rapid temperature increases to cause the deformation. Rather, thermal shock has been more accurately described by others such as Hall, especially in the context of Antarctic environments which might be applicable to this study. This could be a case of needing to be more clear as to what temporal range thermal shock applies to.

Results. Many pages of the results are tedious to read and are presented as detailed descriptions of the exact deformation and temperature changes that the laboratory and field rocks went through. For example, most of Section 4.1 and nearly all of Sections 4.2.3 and 4.2.4 (which encompasses $\sim$ 2 pages of text) could be summarized in a few generalized sentences that describe the results in a chart form. The idea here is to ensure that the reader is guided through the results and their meaning instead of needing to interpret them on their own. As presented, it is not clear what the meaning is of the numerical values presented, other than a description of some parts of the figures.

Discussion. I found the Discussion section lacking in overall applicability to the geomorphological community at large. Typically, discussion sections try to place the context of the study in relationship to existing work. The author does that to some degree,

but mostly only to their own previous publications. I would have liked to see the results compared to those from other researchers working in this field and also to what they think is the likelihood that these results apply to other mountainous settings. In addition, placing the climate change interpretations in the context of others working on similar research in mountainous settings (e.g., Ravenel, Gruber, and others) would also provide the broader applicability that could make this research more impactful. Finally, much of the Discussion (large parts of Sections 5.1 and 5.2, which again span several pages) appear to be a continuation of the intricately detailed results. The Discussion is the section that allow the results to be placed into context, but I did not find this to be the case.

Figures. Whereas this study is clearly complex with bringing together observations from both a laboratory setting and several field sites, I am not sure that there is a need to present all the data for all of the sites in all of the figures. For example, Fig. 4 presents what appears to be similar data for both cracks at RW-1. If this is the case, only one plot could be presented and the other could be moved to the supplementary information. The idea is that if the data does not add to the story, than it does not necessarily need to be presented in the main text. For those scientists interested in using the data, the supplementary information could be consulted. This could also apply to Figs. 6, 7, 8, and 10. As currently presented, the take-home message of the presented results is not clear. On this note, I believe that more detailed captions could assist with describing the salient aspects and findings of the study. On a final note, I found Fig. 11 quite interesting and would recommend that this figure be used to structure the a large part of the Discussion section.

Technical Corrections

L10-12. Suggest rewrite without the numbers (i.e., (1), (2), etc.). This reads awkwardly and the main points of the study are therefore not presented clearly.

L11. Change to "...kinematics of intact rock samples..."

L13. Change to ". . .2935 m in elevation."

L31. Change to "and amplify fracturing by thermo-hydro-mechanical. . ." The sentence requires a verb, so this fixes the issue.

L63. Remove numbers of main points and try summarizing in more typical sentence form.

L63. Change to "Furthermore, I installed. . ."

L126. The use of the C1, C2 terminology should be clarified as these are not shown in Fig. 2.

L161. This section should likely be placed in the Methods.

Fig. 1. It would be helpful to point out that the red parallelogram in (d) is the area shown in (c).

Fig. 4. The forward slash symbol should not be used in the y-axis label, as this appears to represent that rock temperature is being divided by rock top temperature. Better to use a comma to avoid confusion. In addition, consider spelling out some of the abbreviations so that consultation with the caption is not necessary. At a minimum, the rock types could be spelled out, and perhaps the y-axis labels.

Fig. 5. The time frame over which the data is presented is not shown. This is needed to identify the hysteresis effect.

---

## Referee Comment (RC2) · Anonymous Referee #2 · 21 Jan 2021

While not being an expert of the type of research presented in this manuscript, I must say that the description of the methods, data and results looks report-like and seems to be very technical. For non-experts of fracture dynamics related to cooling-warming phases, and maybe also for experts, the research outline is quite tedious. The essential outcome of the laboratory and field tests thus appears to be almost hidden behind these technical descriptions.

Methods, data and results shall be described more concisely, highlighting more the added value of these parallel lab and field tests, for rock mechanics and related hazard understanding. .. that appears a bit as an appendant at the end of the conclusions.

Therefore, I doubt that this paper as such would attract the attention of a wide readership - even though the scientific basis is of a high level.

---

## Author Comment (AC2) · 23 Mar 2021

Reviewer Comments (in black), my response (in blue) and revised manuscript passages (in dark orange)

**Anonymous Referee #2**

While not being an expert of the type of research presented in this manuscript, I must say that the description of the methods, data and results looks report-like and seems to be very technical. For non-experts of fracture dynamics related to cooling-warming phases, and maybe also for experts, the research outline is quite tedious. The essential outcome of the laboratory and field tests thus appears to be almost hidden behind these technical descriptions.

Methods, data and results shall be described more concisely, highlighting more the added value of these parallel lab and field tests, for rock mechanics and related hazard understanding. .. that appears a bit as an appendant at the end of the conclusions. Therefore, I doubt that this paper as such would attract the attention of a wide readership - even though the scientific basis is of a high level.

I like to thank the reviewer for her/his comments. I can understand that my report-like, very technical writing style almost hide my findings. I completely revised the method, result and discussion section following the comments of Reviewer 1. The revised method section starts now with a section describing the laboratory measurements. I added information on mechanical and seismic rock properties that I use for a thermal stress model, I added to the manuscript:

"To understand controlling factors of rock kinematics, I conducted laboratory measurements on airdryed rock samples in a freezing chamber. For this purpose, I collected three approximately 0.4 m long, 0.15 m wide and 0.2 m high rock samples without visible evidence of weathering from talus slopes below rockwalls with lithologies ranging from aplite (AP, RW-1), amphibolite (AM, RW-2) to schistose quartz slate (QS, RW-3; in Fig. 1d). I assume that the rock samples are representative for the rockwall. On each rock sample (Fig. 2), two crackmeters were installed at the top (RD1) and at one side of the sample (RD2) to monitor rock deformation (RD) and rock top temperature (RTT) in 1 min intervals. The two Geokon crackmeters 4420-3 measured RD and automatically correct thermal expansion of the instrument. After temperature correction, the resolution of RD is below 0.00075 mm with an accuracy below 0.003 mm, while RTT is measured with an accuracy of ±0.5°C. Two high-precision Greisinger thermistors connected to a Pt 100 temperature sensor (0.03°C accuracy) were used to monitor rock temperature in the center of the rock sample in 5 cm depth ( $RT_{5cm}$ ) and in 2 cm depth ( $RT_{2cm}$ ). To keep room temperature constant at low levels, the freezing chamber is located in a fridge that keeps the surrounding temperature at 8 to 10 °C. The freezing chamber itself consists of a custom-made Fryka cooler with a temperature-controlled (0.1°C accuracy) ventilation system to enable cooling of samples without thermal layering. The rock samples were cooled down from 10-14 °C to -8/9 °C RT2cm in 7-9 h. Then, I stopped cooling and enhanced a "natural" warming for 15 to 17 h until 8 °C RT2cm was reached. Based on the linear correlation, I derived the thermal expansion coefficient  $\alpha$ :

$$\alpha = \frac{1}{L} * \frac{\Delta RD}{\Delta RT}$$
(1)

with L is crackmeter length,  $\Delta RD$  is rock deformation change and  $\Delta RT$  is rock temperature change.

[revised manuscript text omitted]

Then I present the laboratory tests and validate the observed pattern using the control crackmeters (former dummies). I shortened the description and focus on my main findings: the warming and cooling cycles with different thermal expansion coefficients. I merged Figures 4 and 5 into a new Figure 5 by presenting only the information of the schistose quartz slate sample and RW3. All other subfigures I moved to the supporting information. I show that my control crackmeters show identical behaviour as my lab data. Furthermore, I use the measured thermal expansion coefficients of my lab tests in combination with rock temperature data to model the thermal stress regime. My data clearly shows

that the occurrence of thermal stresses is controlled by snow cover while the magnitude is controlled by topographic controlled insolation (e.g. aspect). I changed the second section of the results to:

"Directly after the start of cooling, several crackmeters revealed a rock expansion, which lasted for a few minutes (initial transition phase; Fig. 5a and Fig. S1c, e-f). Rock samples were cooled down to a RTT between -13.2 °C and -18.4 °C and all crackmeters experienced a negative RD (Fig. 5a-b, Fig. S1-2). Rock temperature in 2 to 5 cm depth was up to 5 to 7 °C higher than RTT during the cooling phase (Fig. 5 and S1-2). Based on  $RT_{5 cm}$  measurements using Eq. (1), thermal coefficient ranged from 5.8 ±0.0  $10^{-1}$  $^{6}$  °C-1 for AM (r2=1) to 7.3 ±0.2 10-6 °C-1 (r2=0.99) for AP and 7.3 ±0.5 10-6 °C-1 (r2=1) for QS during cooling. Nine to 23 min after stopping cooling, several crackmeters at all rock samples (AP RD2, AM RD2, QS RD1 and RD2) experienced a sudden rock deformation ranging from +0.004 mm to +0.0135 mm (transition phase; Fig. 4b, Fig, S2). Despite an increase of RTT from -7.3 to -5 °C, the QS sample and AP RD1 and AM RD1 showed a further rock contraction between -0.0016 mm and -0.0032 mm. The closing behaviour corresponded to the decrease of RT5 cm from -8.2 °C to -9.3 (Fig. 4b). Subsequent warming up to 7.7 or 8 °C RTT resulted in rock expansion (warming phase, Fig. 5a-b and S1-2). This warming phase induced a thermal expansion that corresponds to a thermal expansion coefficient of 7.5 ±0.4 10-  $^{6}$  °C-1 (r2=1) for AP, 7.0 ±0.2 10-6 °C-1 (r2=1) for AM and 7.1 ±1.7 10-6 °C-1 (r2=0.99) for QS. All samples showed a hysteresis effect during warming and cooling cycles (Fig. 4b, Fig. S2), which was amplified using RTT and decreased with further rock temperature depth from RT2 cm to RT5 cm.

Control crackmeters were installed in the field to validate laboratory-observed rock deformation. RTT fluctuated between -15 °C and 10 °C at RW1 in 2017/18, between -20 °C and 25 °C at RW3 from 2016 to 2019 and from -15 °C to 25 °C at RWS from 2017 to 2019 (Fig. 5c, Fig. S3). All control crackmeters showed small daily fluctuations of rock deformation that were even reduced when snow cover occurred. Similar to laboratory experiments, control crackmeters recorded cyclic rock expansion during warming and contrary rock contraction during cooling periods (Fig. 5 d, Fig. S3). The thermal expansion coefficients for cooling and warming based on monthly mean RD and RTT was 6.5  $10^{-6}$  °C-1 for RW1 (r2 = 0.43), 9.4  $10^{-6}$  °C-1 for RW3 (r2 = 0.94) and 9.0  $10^{-6}$  °C-1 for RWS (r2 = 0.9). All control crackmeters showed a hysteresis effect, which was amplified at RW1.

Applying laboratory derived thermal expansion coefficients (Table 1) for warming and cooling to Eq. (2) provides the daily thermal stresses within a rockwall. Daily thermal stresses reflected RST and were increased during snow-free periods in summer and decreased or absent during snow cover periods. Maximum stresses reached up to 2.9-3.0 MPa at RW1, 6.8-7.2 MPa at RW2, 3.1-3.9 MPA at RW3 and 15.4-20.1 MPa at RWS (Fig. 4b-e)."

---

## Author Response (AR1)

**Reviewer Comments (**in black**), my response (**in blue**) and revised manuscript passages** (in dark orange**)**

**Anonymous Referee #1**

*General Comments*

This manuscript presents a highly detailed study of the cooling and heating effects of alpine rocks in a laboratory and field setting. The author should be commended for capturing this type of data set as it is clear that it has the ability to inform on several key attributes of the geomorphological behaviour of high altitude rock walls. The author makes several interesting points and clearly has collected a significant data set to stand behind some important interpretations. The confirmation of hysteretic thermally generated fracture deformation is a very good contribution, as is the observation that the thermal expansion coefficients are different between cooling and warming. In addition, the observations that snow cover (and lack thereof from future climate change) will affect the cyclic thermal effect at different altitudes is an important result in itself, especially since it is clearly shown by the field data. However, overall, I found the structure and presentation of the manuscript requiring significant revision.

At the outset, the Abstract should summarize the main findings of the research. I found these difficult to identify. The Abstract discusses rock fracture, altitudinal dependence, and climate change, but also presents results on shear plane controls, ice segregation effects, and snow cover. Whereas these latter three items are all part of the overall arching theme of rock fracture and altitudinal dependence, I believe that setting up the subject more clearly would assist with the delivery of the major findings. That is, I would suggest text along the lines of: "In this study, I investigate the various altitudinal effects (i.e., thermal cycling, snow cover, ice segregation) on rock fracture and place these in the context of climate change." On this last notion, climate change is discussed in the manuscript, but really only casually at the end and I feel that to tackle this subject, the Discussion should include a more in depth overview of what others have found on this subject. Referring back to the Abstract, I noted that the Conclusions section is almost exactly the same as the Abstract. Please note that the Abstract should provide an overview for readers who have not yet read the paper. The Conclusion summarizes the work for those readers who have finished reading the paper. They should therefore not contain the same identical content. Finally, regarding the overall structure of the manuscript, I found the Results and parts of the Discussion section quite tedious within the overall presentation. Whole sections of the text could be summarized in a table or chart, and it is not necessary to highlight the results of every fracture measurement or temperature reading. I therefore recommend a rewrite of large sections of the text (including the Abstract, Results, Discussion, and Conclusion) so that the results and main findings of the research are more clearly represented. Overall, I found the manuscript did not clearly deliver the apparent intended results of the study.

I like to thank the reviewer for his/her valuable comments. I can understand that the structure and the detailed writing overshadow the main messages of the manuscript. To clarify my intended results, I followed the recommendations of the reviewer and revised the manuscript significantly.

The reviewer criticized the abstract and recommended a revision and a clear setting up of the objectives in the abstract. I completely rewrote the abstract and shortly introduced the geomorphic context: "In alpine environments, tectonic processes, past glaciation and weathering processes fractured rock and prepare or trigger rockfalls, which are important processes of rock slope evolution

and natural hazards." This sentence followed a text that introduce the objectives: "In this study, I quantify thermal- and ice-induced rock and fracture kinematics and place these in the context of their role in producing rockfall and climate change." Then, I shortly introduced the applied techniques and research set up: "I conducted laboratory measurements on intact rock samples, and installed temperature loggers and crackmeters at four rockwalls reaching from 2585 to 2935 m in elevation in the Hungerli Valley, Swiss Alps." and summarized the main findings: "My laboratory data shows that thermal expansion followed three phases of rock kinematics, which resulted in a hysteresis effect. In the field, control crackmeters on intact rock reflected these temperature phases and based on thermal expansion coefficients of these observed phases, I modelled thermal stress. Model results show that thermal stress magnitudes were predominantly below rock strengths. Crackmeters across fractures revealed fracture opening during cooling and reverse closing behaviour during warming on daily scale. Elevation-dependent snow cover controlled the number of daily temperature changes and thermal stresses affecting both intact and fractured rock, while the magnitude is controlled by topographic factors influencing insolation. On seasonal scale, slow ice segregation induced fracture opening can occur within lithology-dependent temperature regimes called frost cracking windows. Shear plane dipping controlled if fractures opened or closed irreversible with time due to thermal-induced block crawling on annual scale." before setting my findings in the larger context of rockfall and climate change: "Climate change will shorten snow duration and increase temperature extremes, therefore, will affect the number and the magnitude of thermal changes and associated stresses. Earlier snowmelt in combination with temperature increase will shift the ice-induced kinematic processes to higher altitudes. In conclusion, climate change will affect and change rock and fracture kinematics and, therefore, change rockfall patterns in Alpine environments." and providing a direction for future work: "Future work should quantify rockfall patterns and link these patterns to climatic drivers. "

To improve the structure and overall tedious representation, I rewrote the result and discussion section. I added a thermal stress model to increase the attention of a wider readership as demanded by Reviewer 2. I changed my style of writing, focused on my key findings, and omitted details that are not necessary to understand the main messages of my manuscript. The aim of my laboratory test was to identify trajectories of thermal-induced rock deformation. The control crackmeters were used to validate these trajectories in the field. I used the thermal expansion coefficients of the laboratory measurements in combination with added rock temperature loggers to drive a thermal stress model. I validate the model results using Schmidt hammer measurements. I added the required information on the research set up in the method section:

"To monitor rock surface temperature (RST), I installed four Maxim iButton DS1922 L temperature loggers with a nominal accuracy of ±0.5 °C in 10 cm deep boreholes following the measurement method by previous studies (e.g. Draebing et al., 2017a; Haberkorn et al., 2015). The loggers recorded RST in 3h intervals between 1 September 2016 and 31 August 2019 (RW1-3) or between 1 September 2017 and 31 August 2019 (RWS), respectively. Due to the logger location in 10 cm depth, I determined snow cover duration using the uniform standard deviation threshold of <0.5 K for positive and negative RST in accordance to Haberkorn et al. (2015). I calculated daily rock temperature warming and cooling cycles $\Delta RT$ and applied the equation from Anderson and Anderson (2012) to model thermal stress $\sigma_{th}$:

$$\sigma_{th} = \frac{\alpha\, E\, \Delta RT}{(1-\nu)}.$$ "

In addition to the more general comments on the result section, the reviewer stated some specific comments:

Results. Many pages of the results are tedious to read and are presented as detailed descriptions of the exact deformation and temperature changes that the laboratory and field rocks went through. For example, most of Section 4.1 and nearly all of Sections 4.2.3 and 4.2.4 (which encompasses _ 2 pages of text) could be summarized in a few generalized sentences that describe the results in a chart form. The idea here is to ensure that the reader is guided through the results and their meaning instead of needing to interpret them on their own. As presented, it is not clear what the meaning is of the numerical values presented, other than a description of some parts of the figures.

and

Figures. Whereas this study is clearly complex with bringing together observations from both a laboratory setting and several field sites, I am not sure that there is a need to present all the data for all of the sites in all of the figures. For example, Fig. 4 presents what appears to be similar data for both cracks at RW-1. If this is the case, only one plot could be presented and the other could be moved to the supplementary information. The idea is that if the data does not add to the story, than it does not necessarily need to be presented in the main text. For those scientists interested in using the data, the supplementary information could be consulted. This could also apply to Figs. 6, 7, 8, and 10. As currently presented, the take-home message of the presented results is not clear. On this note, I believe that more detailed captions could assist with describing the salient aspects and findings of the study.

I restructured the result section into three parts, shortened the result section significantly, merged figures and added more information in the caption to describe the figures. The first part presents the "Meteorological conditions and rock surface temperatures". The section on the rock surface temperatures are new:

[revised manuscript text omitted]

The third part of the result section "Rockwall fracture kinematics" presents the main results of the crackmeters spanning fractures. I significantly shortened the text from two to less than one page. I identified four phases, highlighted these phases in Figure 6 and used these phase to structure the result section:

[Figure]

Figure 6: Crack-top temperature, monthly mean crack-top temperature, crack deformation and monthly mean crack deformation for the crackmeters at (a) RW1, (b) RW2, (c) RW3 and (d) RWS for the period 1 September 2016 to 31 August 2019. Grey rectangles highlight the snow cover period, while numbers indicate the interpreted phases (see Table 4).

I omitted the details of fracture movement from the text and present these details now as a new table structured following the four identified phases:

[revised manuscript text omitted]

The reviewer suggested that subfigures from Figures 7 and 8 can be potentially moved to the supplementary information. Here, I disagree. The differences between the crackmeter kinematics result from different insolation that alters rock temperatures and different insulation from snow cover. These effects become visible when all crackmeters from different altitudes and different aspects are displayed.

I understand that the major pattern of daily temperature magnitudes are visible in one year and I reduced the number of subfigures in former Fig. 10 and moved the former figure to the supplementary information:

[Figure]

Figure 9: Daily crack deformation plotted versus daily crack-top temperature for (a) RW1, (b) RW2, (c) RW3 and (d) RWS in 2018/19. Numbers n indicate the number of daily cycles, NA highlights incomplete measurements during the period due to instrument failure. For results of all periods, see Fig. S5 in the supplementary information.

According to the reviewer: "the Discussion should include a more in depth overview of what others have found on this subject". Furthermore the reviewers suggests a revision: " On a final note, I found Fig. 11 quite interesting and would recommend that this figure be used to structure the large part of the Discussion section."

The reviewer's specific comment include:

"Discussion. I found the Discussion section lacking in overall applicability to the geomorphological community at large. Typically, discussion sections try to place the context of the study in relationship to existing work. The author does that to some degree, but mostly only to their own previous publications. I would have liked to see the results compared to those from other researchers working in this field and also to what they think is the likelihood that these results apply to other mountainous settings. In addition, placing the climate change interpretations in the context of others working on similar research in mountainous settings (e.g., Ravenel, Gruber, and others) would also provide the broader applicability that could make this research more impactful. Finally, much of the Discussion (large parts of Sections 5.1 and 5.2, which again span several pages) appear to be a continuation of the intricately detailed results. The Discussion is the section that allow the results to be placed into context, but I did not find this to be the case."

I can understand the comments and restructured the discussion following the reviewer's recommendation using former Fig. 11 (Fig. 10 in the revised version). I shortened the discussion of the laboratory tests and control crackmeters from two to one pages and omitted the repetitive presentation of the results. In contrast, I focussed more on my main findings, the existence if the three phases: warming, transition and cooling phase:

**„5.1 Cyclic thermal rock deformation**

Cyclic thermal stresses can result in thermal fatigue that breakdown rock and is an important component of mechanical weathering. In my laboratory tests, I can differentiate thermal cycles into three phases: (1) cooling phase, (2) transition phase, (3) warming phase. The (1) cooling phase was characterized by rock contraction with thermal coefficients α (Table 1), which are in the order of previous thermal expansion coefficients for quartz minerals (Siegesmund et al., 2008) or rock samples (Ruedrich et al., 2011; Skinner, 1966). After stopping cooling and enhancing a natural warming of the rock samples, the (2) transition phase started (Fig. 5a-b). Some crackmeters experienced a sudden rock deformation due to rapid rock response to warming, while other crackmeters even on the same rock sample showed further rock contraction (Fig. S1-2). The contraction behaviour corresponded to decreasing $RT_{5\ cm}$, while RTT was already increasing as a response to warmer air temperature. Therefore, the transition phase is characterized by a temperature difference between rock surface (RTT) and rock depth ($RT_{5cm}$), which is a result of the slow speed of heat conduction. While the rock surface (RTT) was warming and rock was expanding, the overall rock kinematics was still controlled by cooling of the rock interior and associated contraction. The (3) warming phase was characterized by rock expansion (Fig. 4 and 5) with thermal expansion coefficients that are slightly different from the cooling (Table 1). Therefore, thermal cycles showed a hysteresis effect, which was previously observed on other lithologies (Ruedrich et al., 2011).

The control crackmeters demonstrated that snow cover controls the occurrence of daily temperature cycles, and associated rock expansion during warming and contrary rock contraction during cooling. Daily temperature changes affected the upper 0.21 to 0.42 m assuming a 12 h temperature cycle and a thermal diffusivity between 1 and 2 mm $s^{-2}$ typical for metamorphic rocks (Vosteen and Schellschmidt, 2003; Cermák and Rybach, 1982). These rock depths correspond to daily temperature cycles in rockwalls observed in previous studies (Gunzburger and Merrien-Soukatchoff, 2011; Anderson, 1998). On annual scale, thermal changes affected 4 to 8 m and, therefore, the entire instrumented rock blocks assuming a half-year cycle and thermal diffusivities typical for metamorphic rocks. Rock temperature boreholes in the Alps suggest heat propagation to similar depths (Phillips et al., 2016a; PERMOS, 2019). Control crackmeters revealed identical phases of thermal-induced rock deformation as laboratory measurements including hysteresis effects. The hysteresis effect was amplified at RW-1, where rock blocks are between 2 and 4 times larger than rock blocks at the other rockwalls (Fig. 3). A larger rock block size results in a longer heat transfer by conduction and increases the transition phase, therefore, I interpret the increase of the hysteresis effect as a result from block size and an effect of scale. The observed differential thermal expansion in combination with different response during transition phase results in mechanical stresses able to breakdown rock that will be discussed in Chapter 5.3."

In section 2 of the discussion section, I focus on the fracture kinematics and put my results in the context of previous studies. I also shortened this section by 50%. Most studies focused on sheeting joints where progressive rock movement can trigger rockfall in terms of toppling or wedge failures. However, not all moving blocks are located in steep or overhanging rockwalls but these blocks can progressively move by identical processes as demonstrated by my data and the thermal- and ice-induced movement can also result in rockfall.

"In overhanging steep rock cliffs, thermal stresses can propagate fractures and trigger rockfall (Collins and Stock, 2016; Stock et al., 2012). Exfoliation joints warm differentially (Guerin et al., 2019) and thermal bowing results in fracture propagation parallel to the rockwall (Collins and Stock, 2016). In fractured rock, daily and annual thermal cycles result in rock contraction with fracture widening during cooling and rock expansion during warming with fracture closing (Fig. 11b; Cooper and Simmons, 1977; Draebing et al., 2017b). My data showed that crack opening and closing occurred on a daily scale (Fig. 8). The magnitude of this crack deformation depends on the magnitude of temperature change and was increased at RWS and RW3, which experienced much higher daily temperature changes (Fig. 9). The data demonstrated that the number of daily temperature changes per year (Fig. 9 and Fig. S5) is controlled by the duration of snow cover. Snow cover insulates the ground (Zhang, 2005) and decreases daily thermal changes (Fig. 10; Luetschg and Haeberli, 2005; Draebing et al., 2017b), therefore, rockwalls at lower elevations (e.g. RW3) or south-exposed rockwalls (e.g. RWS) experienced a shorter snow cover duration (Table 2) and a higher frequency of thermal cycles than higher-elevated rockwalls with longer snow cover (e.g. RW2-3).

On seasonal scale, several crackmeters experienced a slow crack opening during negative CTT that suggest the influence of ice (Fig. 11c). Matsuoka (2001, 2008) observed rapid crack opening during freeze-thaw cycles in spring or autumn that suggest volumetric expansion as the trigger of crack opening. In contrast, I observed a slow crack opening lasting between 1.5 to 2 months at RW2 in 2017 (Fig. 6b and Fig. 7c-d), 3 to 3.5 months at Crack-2 at RWS in 2018 (Fig. 6d and Fig. 7h), and 5 to 5.5 months at Crack-2 at RW1 in 2017 and 2019 (Fig. 6a and Fig. 7b). The opening occurred in a CTT range between -5 to -3.3°C at RW2, between -8 and -3.6°C at RWS and -9 to -1°C at RW1. These temperature ranges are within the frost cracking window from -8 to -3 °C suggested by Anderson (1998) or in the range of laboratory observed frost cracking windows of aplite, amphibolite and schist quartz slate (Draebing and Krautblatter, 2019). The temporal trajectories of the affected cracks showed that crack opening occurred during warming within the negative temperature range (Fig. 7b-d. h), therefore, a purely thermal response of the rock would result in rock expansion and crack closing. In addition, a transition phase as observed in the laboratory or at control crackmeters can be excluded, thus, the transition phase is associated with temperature changes from warming to cooling or reverse, which were absent during these crack opening phases. Therefore, I interpret the observed crack opening as a result of ice segregation and subsequent closing because of ice relaxation. Cracks can infill with ice by refreezing of meltwater, ice infill growth slowly with time by cryosuction (Weber et al., 2018; Draebing et al., 2017b) and build up ice pressure and stresses in a subcritical range (Draebing and Krautblatter, 2019), which induce slow fracture opening (Draebing and Krautblatter, 2019; Draebing et al., 2017b).

[revised manuscript text omitted]

Then I focus on cryogenic fracture displacement and put my results into the context of the frost cracking window and potential altitudinal shifts of this window due to climate change:

"Cryogenic processes can cause fracture movement (Draebing et al., 2017b) and produces subcritical stresses that are below tensile strengths of rocks (Draebing and Krautblatter, 2019). These stresses occur in a wide temperature range between -15 and -1°C according to field measurements (Girard et al., 2013; Amitrano et al., 2012), numerical models (Walder and Hallet, 1985) and laboratory tests (Draebing and Krautblatter, 2019). My data showed the occurrence of cryogenic induced opening and closing at several crackmeters (Fig. 7b-d, 7h). The slow opening in combination with the temperature regime suggests that ice segregation is the driving process behind the observed crack opening. This opening can progressively weaken rockwall strength and potentially triggers rock slope failure as observed at Piz Kesch in winter 2014 (Phillips et al., 2016b). The length of the temperature window enabling ice segregation increases with altitude but is modulated by insulating snow cover. Climate change will shorten the snow duration by an earlier meltout date (Bender et al., 2020) and increase air and ground temperatures (Bender et al., 2020; Gobiet et al., 2014), therefore, the time period of temperature windows enabling ice segregation will decrease. This will affect especially rockwalls located at lower elevations, therefore, cryogenic processes and triggered rockfall will be shifted to higher elevations. At higher-elevations, the climate-changed induced changes of the temperature regime will also affect permafrost rockwalls, decrease rockwall stability (Draebing et al., 2014; Krautblatter et al., 2013) and increase rockfall activity due to increased thawing (Ravanel et al., 2010; Ravanel and Deline, 2010) amplified by temperature extremes (Gruber et al., 2004b; Ravanel et al., 2017)."

Finally, I focus on observed rock creeping, which is controlled by geology and put my results into a wider context of rock slope movement or deep-seated gravitational slope deformations:

"Plastic deformation of fractures contributes to the preparation of rockfalls (Gunzburger et al., 2005). Previous studies demonstrated that thermal-induced crack deformation causes irreversible displacement of rock blocks (Weber et al., 2017; Hasler et al., 2012). Gunzburger et al. (2005) and do Amaral Vargas et al. (2013) suggested that thermal changes can cause rock creep along shear planes. My data demonstrates that rock blocks creep with a direction depending on shear plane dipping. Dipping out of the rockwall results in creeping that increases fracture opening (RW1, Fig. 10d) and can trigger rockslide processes. In contrast, dipping into the rockwall causes fracture closing (RW3, RWS; Fig. 10e). My observed blocks will not be released as rockfall, however, the observed mechanism can trigger a wedge failure in more steeper and overhanging rockwalls that are more susceptible to rockfall (Matasci et al., 2017). Several studies indicated that thermal-induced stresses can affect rock slopes up to 100 m depth (Gischig et al., 2011b) and affect deep-seated gravitational slope deformations (Gischig et al., 2011a; Rouyet et al., 2017; Watson et al., 2004). Therefore, thermal changes can decrease rock slope stability with time (preparatory factor) and potentially cause landsliding (triggering factor)."

The reviewer commended that: "Conclusions section is almost exactly the same as the Abstract. Please note that the Abstract should provide an overview for readers who have not yet read the paper. The Conclusion summarizes the work for those readers who have finished reading the paper. They should therefore not contain the same identical content."

I rewrote the conclusion section and omitted all sentences on the set up and methods that the reader already knows and focused on the main findings:

"Thermal-induced rock deformation can cause fracture movement and thermal stresses able to breakdown rock. Cyclic thermal rock deformation follows a cooling and warming phase with different thermal expansion coefficients. The transition between these phases are characterised by both rock contraction and expansion and thermal cycles show a hysteresis effect observed in the laboratory and the field. Snow cover controls the number of daily temperature changes, while topographic factors influencing insolation (e.g. aspect) controls the magnitude of temperature changes. Months-long slow crack opening occurred within a temperature range between -9 and -1°C suggested that ice segregation caused slow fracture opening reversed by ice relaxation. Depending on the dipping of shear planes, thermal changes on annual scale causes rock creeping that widen or shorten fracture aperture. Cycles of thermal- and ice-induced crack opening and closing can progressively weaken rockwalls and trigger rockfall. Climate change will shorten snow duration and increase temperature extremes, therefore, will affect the number and the magnitude of thermal changes and associated stresses. However, an earlier snowmelt in combination with temperature increase will decrease the occurrence of ice segregation and will shift the frost favouring temperature regime to higher altitudes. Therefore, climate change will change the frequency, magnitude and the location of thermal and ice-induced stresses and will change the spatial variation of rockfall in alpine environments in the future."

**Specific Comments**

Dummy Crackmeters. The use of the word "Dummy" should probably be changed to "Control". Dummy implies that the device does not provide any useful information. Regarding the devices themselves, they are not quite control crackmeters either, however, since they are still measuring rock deformation, just not across a fracture. Thus, whereas they provide some guidance for understanding the fracture measurements, there use should be put into context that they are better for understanding the nonfractured behaviour of the rock mass. Also, since the crackmeters were temperature corrected (L113), it was unclear how the controls were used for verification of the measured signals.

Thank you for this comment. I changed the wording from dummy to control crackmeter. The aim of the control crackmeters was to validate the findings of the laboratory tests. To clarify this, I added the information to the objectives at the end of the introduction: "Furthermore, I installed control crackmeters and rock temperature loggers on intact rocks at four rockwalls reaching from 2585 to 2935 m in elevation to validate laboratory-derived rock deformation behaviour and to model thermal stresses." and in the method section: "To validate rock deformation (RD) observed in the laboratory, one crackmeter was used as a control crackmeter and was fixed on intact bedrock without any cracks (Fig. 3)."

Thermal Shock. The term thermal shock is used to describe rapid deformation and fracture observed in several previous studies (L265), however my understanding is that those studies did not depend on rapid temperature increases to cause the deformation. Rather, thermal shock has been more accurately described by others such as Hall, especially in the context of Antarctic environments which might be applicable to this study. This could be a case of needing to be more clear as to what temporal range thermal shock applies to.

The reviewer is right and thermal shock is defined differently. Hall and Thorn review the process of thermal shock. They state that thermal stress is generated by sudden temperature changes that exceed the strength of material. The temperature change can be warming or cooling. The authors also

summarize different thresholds that depend on temperature changes per time and are all larger than $1°C\ min^{-1}$. Therefore, I conclude that my observed rock deformation is not associated to thermal shock and more associated to temperature gradients between the rock surface and the rock interior. I omitted the term thermal shock and changed the text in the discussion to: "After stopping cooling and enhancing a natural warming of the rock samples, the (2) transition phase started (Fig. 5a-b). Some crackmeters experienced a sudden rock deformation due to rapid rock response to warming, while other crackmeters even on the same rock sample showed further rock contraction (Fig. S1-2). The contraction behaviour corresponded to decreasing RT5 cm, while RTT was already increasing as a response to warmer air temperature. Therefore, the transition phase is characterized by a temperature difference between rock surface (RTT) and rock depth (RT5cm), which is a result of the slow speed of heat conduction. While the rock surface (RTT) was warming and rock was expanding, the overall rock kinematics was still controlled by cooling of the rock interior and associated contraction."

***Technical Corrections***

L10-12. Suggest rewrite without the numbers (i.e., (1), (2), etc.). This reads awkwardly and the main points of the study are therefore not presented clearly.

Done.

L11. Change to ". . .kinematics of intact rock samples. . ."

I rewrote the text to: "In this study, I quantify thermal- and ice-induced rock and fracture kinematics and place these in the context of their role in producing rockfall and climate change."

L13. Change to ". . .2935 m in elevation."

Done.

L31. Change to "and amplify fracturing by thermo-hydro-mechanical. . ." The sentence requires a verb, so this fixes the issue.

Done.

L63. Remove numbers of main points and try summarizing in more typical sentence form.

Done.

L63. Change to "Furthermore, I installed. . ."

Done.

L126. The use of the C1, C2 terminology should be clarified as these are not shown in Fig. 2.

I changed the labelling of laboratory crackmeters to RD1 and RD2 in the text, which is consistent to Fig.2.

L161. This section should likely be placed in the Methods.

I followed the advice and incorporated the section into the method section:" To monitor fracture movement in the field, I installed three 0.4 m long Geokon Vibrating-Wire Crackmeters 4420-1-50 with resolution of 0.0125 mm and accuracy of 0.05 mm at RW-1 to RW-3 in 2016 (Fig. 3a-c). Instrumented rockwalls range from 2585 m (RW-3), 2672 m (RW-2) to 2935 m (RW-1) and are exposed in northwest (NW) to northeast (NE) direction (Fig. 1; Table 2). Rock strength of each rockwall was measured using a N-type Schmidthammer following Selby (1980, Table 3). At RW-1, a 3.5 m long, 1.9 m wide and 2.1 m high aplite block was monitored (Fig. 3a). A slickenslide on top of the block indicates former movement of a previously above-laying block. The monitored block slides on a 20 to 40° inclined shear plane (J3 in Fig. 3a) into the valley with an identical angle than the slickenslide. The block is separated by a 10 to 70 mm wide crack of joint set 1 (J1) from a second 2.5 m long and 2.6 m high block. At RW-2, three blocks were monitored which are incorporated in a heavily fractured rockwall consisting of amphibolite. All monitored cracks possess an aperture between 1 and 2 mm (Table 1) and are dipping at 50° out of the rockwall (J2), however, the blocks are buttressed by adjacent blocks and the talus slope (Fig. 3b). RW-3 consists of schistose quartz slate and blocks are 0.3 to 1.5 m wide (Fig. 3c). Blocks are separated by 81° inclined and 30 to 50 mm wide cracks (J2), which were monitored. The schist cleavage is dipping at 22° into the rockwall.

To identify effects of different exposition, three more crackmeters were installed at RW-S at 2723 m in 2017 (Fig. 3d). RW-S is heavily fractured, which results in the occurrence of a high number of joint sets (Fig. 3d). The monitored blocks dip with an angle of 41° degrees into the rockwall (H2) and crack aperture range from 7.5 to 20 mm at Crack-2 to 45 to 135 mm at Crack-1 (Table 2)."

Fig. 1. It would be helpful to point out that the red parallelogram in (d) is the area shown in (c).

I changed the figure caption and added one sentence: "The red parallelogram highlights the photo extend of (c)."

Fig. 4. The forward slash symbol should not be used in the y-axis label, as this appears to represent that rock temperature is being divided by rock top temperature. Better to use a comma to avoid confusion. In addition, consider spelling out some of the abbreviations so that consultation with the caption is not necessary. At a minimum, the rock types could be spelled out, and perhaps the y-axis labels.

AND

Fig. 5. The time frame over which the data is presented is not shown. This is needed to identify the hysteresis effect.

I followed the reviewer's advice and changed the "RT/RTT" to "RT, RTT" in all figures. I also added a time frame and arrow indicating warming and cooling to Figure 5 and Figure S2 in the supplementary information.

[Figure]

Figure 5: Laboratory crackmeter measurements of rock samples. (a) Rock-top temperature (RTT), rock temperature (RT) and rock deformation (RD) of crackmeter 1 and 2 plotted versus time for the schistose quartz slate sample (QS). (b) Rock deformation plotted versus rock-top temperature or rock temperature for crackmeter 2 of QS. Numbers indicate the timing of the beginning of the cooling period (black dot), end of the cooling period (black rectangle) and beginning of the warming period (black triangle). Arrows highlight the temporal trajectory of cooling (blue) and warming (red). Results of all laboratory measurements are shown in Fig. S1 and S2 in the supplementary information. Field measurements of the control crackmeter at RW3. (c) RTT (red line) and monthly mean RTT (dark red line), RD (blue line) and monthly mean RD (dark blue line) plotted versus time for the period from 1 September 2016 to 31 August 2019. Grey rectangles highlight the occurrence of snow cover. (d) Monthly mean rock deformation plotted versus monthly mean RTT. Red rectangles indicate start, black dots first day of a month, red dots the beginning of new measurement period and red triangles the end of measurements. Colour of graphs indicate data from 2016/17 (light blue), from 2017/18 (blue) and 2018/19 (green). For results of RW1 and RWS, see Fig. S3 in the supplementary information.

**Anonymous Referee #2**

While not being an expert of the type of research presented in this manuscript, I must say that the description of the methods, data and results looks report-like and seems to be very technical. For non-experts of fracture dynamics related to cooling-warming phases, and maybe also for experts, the research outline is quite tedious. The essential outcome of the laboratory and field tests thus appears to be almost hidden behind these technical descriptions.

Methods, data and results shall be described more concisely, highlighting more the added value of these parallel lab and field tests, for rock mechanics and related hazard understanding. .. that appears a bit as an appendant at the end of the conclusions. Therefore, I doubt that this paper as such would attract the attention of a wide readership - even though the scientific basis is of a high level.

I like to thank the reviewer for her/his comments. I can understand that my report-like, very technical writing style almost hide my findings. I completely revised the method, result and discussion section following the comments of Reviewer 1. The revised method section starts now with a section describing the laboratory measurements. I added information on mechanical and seismic rock properties that I use for a thermal stress model, I added to the manuscript:

[revised manuscript text omitted]

Then I present the laboratory tests and validate the observed pattern using the control crackmeters (former dummies). I shortened the description and focus on my main findings: the warming and cooling cycles with different thermal expansion coefficients. I merged Figures 4 and 5 into a new Figure 5 by presenting only the information of the schistose quartz slate sample and RW3. All other subfigures I moved to the supporting information. I show that my control crackmeters show identical behaviour as my lab data. Furthermore, I use the measured thermal expansion coefficients of my lab tests in combination with rock temperature data to model the thermal stress regime. My data clearly shows

that the occurrence of thermal stresses is controlled by snow cover while the magnitude is controlled by topographic controlled insolation (e.g. aspect). I changed the second section of the results to:

[revised manuscript text omitted]

I shortened the results section on fracture kinematic by 50 % and focused concisely on the patterns. I identified four phases that I highlighted in Figure 6:

[Figure]

**Figure 6: Crack-top temperature, monthly mean crack-top temperature, crack deformation and monthly mean crack deformation for the crackmeters at (a) RW1, (b) RW2, (c) RW3 and (d) RWS for the period 1 September 2016 to 31 August 2019. Grey rectangles highlight the snow cover period, while numbers indicate the interpreted phases (see Table 4).**

I added a new Table 4 that summarizes the kinematic patterns of these phases as well as presents annual and overall deformation:

**Table 4: Quantified rock fracture kinematic phases, annual and overall crack deformation in mm. Crack deformation (CD) is differentiated in crack closing (-), crack opening (+) and no CD (±).**

| Phases/ Cumulative crack deformation | RW1 Crack-1 | RW1 Crack-2 | RW2 Crack-1 | RW2 Crack-2 | RW3 Crack-1 | RW3 Crack-2 | RWS Crack-1 | RWS Crack-2 |
|---|---|---|---|---|---|---|---|---|
| *Phase 1: cooling period* | | | | | | | | |
| 2016-17 | +0.53 | +0.09 | +0.13 | +0.09 | +0.17 | -0.28 | | |
| 2017-18 | +0.22 | +0.04 | | | +0.33 | +0.01 | -0.20 | -0.12 |
| 2018-19 | +0.32 | +0.08 | | | +0.15 | ±0.00 | -0.18 | -0.10 |
| *Phase 2* | | | | | | | | |
| 2017 | -0.05 | +0.07 | +0.02 | +0.11 | | | | |
| 2018 | +0.02 | +0.03 | | | | | | +0.16 |
| 2019 | -0.08 | -0.01 | | | | | | |
| *Phase 3* | | | | | | | | |
| 2017 | -0.31 | -0.10 | -0.15 | -0.29 | | | | |
| 2018 | -0.18 | -0.07 | | | | | | -0.13 |
| 2019 | -0.12 | +0.03 | | | | | | |
| *Phase 4: warming period* | | | | | | | | |
| 2017 | +0.07 | +0.05 | | | -0.31 | +0.04 | | |
| 2018 | +0.06 | +0.12 | | | -0.33 | -0.05 | +0.14 | ±0.00 |
| 2019 | +0.06 | +0.11 | | | -0.13 | -0.04 | +0.13 | +0.08 |
| *cumulative annual CD* | | | | | | | | |
| 2016/17 | +0.24 | +0.11 | ±0.00 | -0.10 | -0.09 | -0.24 | | |
| 2017/18 | +0.12 | +0.12 | | | -0.02 | -0.08 | -0.06 | -0.09 |
| 2018/19 | +0.20 | +0.22 | | | -0.13 | +0.07 | -0.03 | -0.04 |
| ***overall CD*** | ***+0.56*** | ***+0.45*** | | | ***-0.24*** | ***-0.25*** | ***-0.09*** | ***-0.13*** |

I restructured my result section and focussed on the identified phases:

"Crackmeters showed a decrease of snow duration with decreasing elevation. Crackmeters at RW1 located at 2935 m experienced 192 to 223 days of snow cover per year (Fig. 6). The number of snow-covered days was reduced to 69 and 73 days in 2016/17 at RW2 at 2672 m. Snow load damaged the equipment in the following years, which resulted in a data gap and incomplete measurement of snow cover duration. RW-3 at 2585 m showed between zero and nine days of snow cover and snow cover was limited to two days at Crack-2 in 2017/18 at the south-facing rockwall RWS.

All crackmeters experienced a cooling period (Phase 1 in Fig. 6) that ranged from September until mid-December to Mid-February. At RW1, the onset of snow cover controlled the end of the cooling period, only in 2016/17 cooling continued under snow cover. The cooling period was characterized by a crack opening (Table 4), which was between 0.22 and 0.53 mm at Crack-1 and between 0.04 and 0.09 mm at Crack-2 at RW1. At RW2, cracks opened between 0.09 mm at Crack-2 and 0.13 mm at Crack-1, while Crack-1 at RW3 revealed a crack opening between 0.15 and 0.33 mm. In contrast, Crack-2 at RW3 experienced a diverse crack deformation ranging from crack closing between 0.28 mm to 0.01 mm crack opening. At RWS, the cooling period was characterized by crack closing between 0.18 and 0.20 mm at Crack-1 and between 0.1 and 0.12 mm at Crack-2.

At RW1 and RW2, crackmeters experienced a slow warming below snow cover (Phase 2 in Fig. 6), which resulted in either crack opening or crack closing (Table 4). Crackmeters at RW1 experienced a crack deformation ranging from closing of 0.08 mm to crack opening of 0.07 mm. At RW2, cracks experienced a crack opening between 0.02 and 0.11 mm. At RWS, snow cover was absent during warming, however, Crack-2 experienced an opening of 0.16 mm, which was reversed by 0.13 mm during enhanced warming in 2017. RW1 and RW2 showed a period of predominantly crack closing during enhanced warming until snow cover completely melted (Phase 3 in Fig. 6). Crack-1 at RW1 experienced between 0.12 to 0.31 mm crack closing, while Crack-2 showed a diverse crack deformation ranging between 0.03 mm opening and 0.10 mm closing. At RW2, the crackmeters revealed crack closing between 0.15 and 0.29 mm.

All crackmeters experienced a warming period (Phase 4 in Fig. 6), which is characterized by both crack closing and crack opening (Table 4). At RW1, crackmeters revealed a crack opening between 0.05 and 0.12 mm. In contrast, RW3 showed a crack closing between 0.13 and 0.33 mm at Crack-1 and a crack deformation behaviour ranging from 0.07 crack opening to 0.24 mm crack closing at Crack-2. At RWS, the warming period is associated with crack opening between 0.00 and 0.14 mm. In the 3-year period, RW1 experienced a crack opening between 0.45 and 0.51 mm. In contrast, RW3 showed an overall crack closing between 0.24 and 0.25 mm. Data at RW2 was limited to 2016/17 and cracks experienced no change or 0.1 mm closing. RWS was characterized by crack closing in each year with a cumulative closing between 0.03 and 0.09 mm.

On a daily scale, cooling resulted in crack opening due to contraction of two rock blocks and warming in crack closing due to expansion of two rocks (Fig. 8). The peak of opening and closing between individual crackmeters can be different. Crackmeters at RW1 experienced daily CTT fluctuations in the range below 10°C and daily CD below 0.1 mm (Fig. 9) during snow-free periods. The fluctuations were increased up to 16°C and 0.11 mm at RW2, 23 °C and 0.26 mm at RW3 as well as 21°C and 0.32 mm at RW-S."

I completely revised the discussion section. I omitted the repetitive description of results and focussed more on my identified patterns. Following the recommendations of Reviewer 1, I structured the discussion along Figure 11 (now Fig. 10). In the first section, I focus on "5.1 Cyclic thermal rock deformation". My major findings disused in this section are:

**My laboratory data show that warming and cooling on sample scale shows a trajectory with different thermal expansion coefficients and a hysteresis effect**. My data can be used to drive a thermal stress model, as added to the revised manuscript. Thermal stresses can develop that can concentrate at fracture and widen these fractures. **My control crackmeters (former dummies) demonstrate that identical trajectories can be observed on annual scale**. I revised the first discussion section to: "Cyclic thermal stresses can result in thermal fatigue that breakdown rock and is an important component of mechanical weathering. In my laboratory tests, I can differentiate thermal cycles into three phases: (1) cooling phase, (2) transition phase, (3) warming phase. The (1) cooling phase was characterized by rock contraction with thermal coefficients α (Table 1), which are in the order of previous thermal expansion coefficients for quartz minerals (Siegesmund et al., 2008) or rock samples (Ruedrich et al., 2011; Skinner, 1966). After stopping cooling and enhancing a natural warming of the rock samples, the (2) transition phase started (Fig. 5a-b). Some crackmeters experienced a sudden rock deformation due to rapid rock response to warming, while other crackmeters even on the same rock sample showed further rock contraction (Fig. S1-2). The contraction behaviour corresponded to decreasing $RT_{5\ cm}$, while RTT was already increasing as a response to warmer air temperature. Therefore, the transition phase is characterized by a temperature difference between rock surface (RTT) and rock depth ($RT_{5cm}$),

which is a result of the slow speed of heat conduction. While the rock surface (RTT) was warming and rock was expanding, the overall rock kinematics was still controlled by cooling of the rock interior and associated contraction. The (3) warming phase was characterized by rock expansion (Fig. 4 and 5) with thermal expansion coefficients that are slightly different from the cooling (Table 1). Therefore, thermal cycles showed a hysteresis effect, which was previously observed on other lithologies (Ruedrich et al., 2011).

The control crackmeters demonstrated that snow cover controls the occurrence of daily temperature cycles, and associated rock expansion during warming and contrary rock contraction during cooling. Daily temperature changes affected the upper 0.21 to 0.42 m assuming a 12 h temperature cycle and a thermal diffusivity between 1 and 2 mm s$^{-2}$ typical for metamorphic rocks (Vosteen and Schellschmidt, 2003; Cermák and Rybach, 1982). These rock depths correspond to daily temperature cycles in rockwalls observed in previous studies (Gunzburger and Merrien-Soukatchoff, 2011; Anderson, 1998). On annual scale, thermal changes affected 4 to 8 m and, therefore, the entire instrumented rock blocks assuming a half-year cycle and thermal diffusivities typical for metamorphic rocks. Rock temperature boreholes in the Alps suggest heat propagation to similar depths (Phillips et al., 2016a; PERMOS, 2019). Control crackmeters revealed identical phases of thermal-induced rock deformation as laboratory measurements including hysteresis effects. The hysteresis effect was amplified at RW-1, where rock blocks are between 2 and 4 times larger than rock blocks at the other rockwalls (Fig. 3). A larger rock block size results in a longer heat transfer by conduction and increases the transition phase, therefore, I interpret the increase of the hysteresis effect as a result from block size and an effect of scale. The observed differential thermal expansion in combination with different response during transition phase results in mechanical stresses able to breakdown rock that will be discussed in Chapter 5.3."

In the second part of the discussion section, I focus on the identified fracture kinematic patterns. My major findings are: **On daily scale, my data showed thermal-induced crack opening and closing. The occurrence is controlled by snow and the magnitude by insolation.** Furthermore my data suggest **the occurrence of slow ice-induced fracture opening that takes place in a temperature regime between -9 and -1° C very similar to the frost cracking window on seasonal scale**. Ice-induced fracture opening is assumed to be a preparing factor and also a trigger of rockfall, however, this process was rarely observed and only one study previously quantified this process. **On annual scale, my data demonstrated an irreversible rock movement due to rock creep**. Several field studies suggested that blocks can creep and that this creep can be irreversible. Laboratory studies confirmed this process, up to my knowledge this process was not observed or quantified before. I changed the section on fracture kinematics to: "In overhanging steep rock cliffs, thermal stresses can propagate fractures and trigger rockfall (Collins and Stock, 2016; Stock et al., 2012). Exfoliation joints warm differentially (Guerin et al., 2019) and thermal bowing results in fracture propagation parallel to the rockwall (Collins and Stock, 2016). In fractured rock, daily and annual thermal cycles result in rock contraction with fracture widening during cooling and rock expansion during warming with fracture closing (Fig. 11b; Cooper and Simmons, 1977; Draebing et al., 2017b). My data showed that crack opening and closing occurred on a daily scale (Fig. 8). The magnitude of this crack deformation depends on the magnitude of temperature change and was increased at RWS and RW3, which experienced much higher daily temperature changes (Fig. 9). The data demonstrated that the number of daily temperature changes per year (Fig. 9 and Fig. S5) is controlled by the duration of snow cover. Snow cover insulates the ground (Zhang, 2005) and decreases daily thermal changes (Fig. 10; Luetschg and Haeberli, 2005; Draebing et al., 2017b), therefore, rockwalls at lower elevations (e.g. RW3) or south-exposed rockwalls

(e.g. RWS) experienced a shorter snow cover duration (Table 2) and a higher frequency of thermal cycles than higher-elevated rockwalls with longer snow cover (e.g. RW2-3).

On seasonal scale, several crackmeters experienced a slow crack opening during negative CTT that suggest the influence of ice (Fig. 11c). Matsuoka (2001, 2008) observed rapid crack opening during freeze-thaw cycles in spring or autumn that suggest volumetric expansion as the trigger of crack opening. In contrast, I observed a slow crack opening lasting between 1.5 to 2 months at RW2 in 2017 (Fig. 6b and Fig. 7c-d), 3 to 3.5 months at Crack-2 at RWS in 2018 (Fig. 6d and Fig. 7h), and 5 to 5.5 months at Crack-2 at RW1 in 2017 and 2019 (Fig. 6a and Fig. 7b). The opening occurred in a CTT range between -5 to -3.3°C at RW2, between -8 and -3.6°C at RWS and -9 to -1°C at RW1. These temperature ranges are within the frost cracking window from -8 to -3 °C suggested by Anderson (1998) or in the range of laboratory observed frost cracking windows of aplite, amphibolite and schist quartz slate (Draebing and Krautblatter, 2019). The temporal trajectories of the affected cracks showed that crack opening occurred during warming within the negative temperature range (Fig. 7b-d. h), therefore, a purely thermal response of the rock would result in rock expansion and crack closing. In addition, a transition phase as observed in the laboratory or at control crackmeters can be excluded, thus, the transition phase is associated with temperature changes from warming to cooling or reverse, which were absent during these crack opening phases. Therefore, I interpret the observed crack opening as a result of ice segregation and subsequent closing because of ice relaxation. Cracks can infill with ice by refreezing of meltwater, ice infill growth slowly with time by cryosuction (Weber et al., 2018; Draebing et al., 2017b) and build up ice pressure and stresses in a subcritical range (Draebing and Krautblatter, 2019), which induce slow fracture opening (Draebing and Krautblatter, 2019; Draebing et al., 2017b).

[revised manuscript text omitted]

The reviewer doubts that this study is of interest to a wider readership and I can only disagree. This study quantifies rock and fracture kinematics and partially stresses. It links these stresses to climate and discusses potential effects of climate change that can change rockfall in mountainous areas in the future. I admit that the first version of the manuscript was tedious and hid the major findings, however, I think these findings are presented in a concise and improved way in the revised version. Furthermore, I think the findings are of wider interest. I quantified climatic drivers of rockfall, which explain rockfall occurrences in Alpine environments and can be quantified by rockfall data in future studies.

---

## Author Response (AR2)

**Reviewer Comments (**in black**), my response (**in blue**) and revised manuscript passages** (in dark orange**)**

**Change of 'Altitude' to 'Elevation'**

After my last published paper, Tim McVicar wrote me last week and explained me that I misuse the term altitude. I read his paper (McVicar & Körner, 2012) and understand my mistake. In this paper, I repeated my mistake. Therefore, I change the term 'altitude' into 'elevation' within this revision.

**Anonymous Referee #3**

**Minor revision** - the Abstract states that 'Future work should quantify rockfall patterns and link these patterns to climatic drivers'. However, this recommendation is not reiterated in the Conclusion, and, as far as I can see, it is not stated clearly or developed in more detail in the Discussion. As written in the Abstract this suggestion seems like an afterthought, and I'd like to see more development/detail included in the Discussion to this effect.

Thank you for this valuable comment. To reiterate the statement, I included two new sentences in the discussion. In the section on thermal stresses I state that "Consequently, thermal stresses will play a more important role in preparing and triggering rockfall in the future." I add a follow-up sentence "Future work should quantify rockfall and link rockfall patterns to modelled or measured thermal stresses". After discussing the influence of climate change on cryogenic stresses, I add a statement "Rockfall should be quantified and linked to climatic drivers to predict the effects of climate change on rockfall more accurately." In the conclusion, I close the loop and state: "To quantify rockfall patterns and link rockfall patterns to climatic drivers should be the objective of future work".

**Technical corrections**:

L9 – 'fractured' should be 'fracture' Done.

L18 – correct to 'on daily time scales' Done.

L19 – correct to 'On a seasonal scale…' Done.

L20 – correct to '…ice segregation-induced fracture opening…' Done.

L21 – change 'irreversible' to 'irreversibly' Done.

L23 – change to 'on an annual scale' Done.

L22 – correct to '..and increase temperature extremes and will, therefore, affect the number…' Done.

L25 - I think 'Alpine' should be 'alpine'. Your study is 'Alpine'-focused, but the implications of your findings go beyond the Alps, therefore suggest you use lowercase 'alpine', as per line 9. Also, L30 – I think this should be 'alpine' too. Check remainder of manuscript – there are numerous instances of use that need checking.

I followed the recommendation of the reviewer an changed 'Alpine' to 'alpine' in the manuscript.

The order of in-text citations needs checking and correcting throughout so that where more than one citation occurs, these are ordered by date (earliest first). There's no logic to your system at present. I revised the citations and in-text citations are now ordered by date starting from old to young.

L52 – correct to 'expansion-induced' Done.

L84 – 'air-dried' Done.

L234 – 'break down'. Also L370. Done.

L290 – correct to 'On an annual time scale' Done.

L333 – correct to '…are not acting in isolation and other…' Done.

L335 – correct to '…As soon as cracks exist, daily…' Done.

L338 – correct to '…changes in snow duration…' Done.

L375 – change to '…suggests that ice segregation caused slow fracture opening, which was reversed by ice relaxation.' Done.

(note - there are more typos and minor grammatical errors than I've listed here, but I'll defer to the copy-editing team to pick these up and correct them in due course)

Thanks for correcting the typos. I found some more an hope that the text is now (more) correct.